# INFER : Learning Implicit Neural Frequency Response Fields for Confined Car Cabin

## Abstract

Accurate modeling of spatial acoustics is critical for immersive and intelligible audio in confined, resonant environments such as car cabins. Current tuning methods are manual, hardware-intensive, and static, failing to account for frequency selective behaviors and dynamic changes like passenger presence or seat adjustments. To address this issue, we propose **INFER** (**I**mplicit **N**eural **Fre**quency **R**esponse fields), a frequency-domain neural framework that is jointly conditioned on source and receiver positions, orientations to directly learn complex-valued frequency response fields inside confined, resonant environments like car cabins. We introduce three key innovations over current neural acoustic modeling methods: (1) an end-to-end neural frequency response field that directly learns frequency-specific attenuation in 3D space; (2) perceptual and hardware-aware spectral supervision that emphasizes critical auditory frequency bands and deemphasizes unstable crossover regions; and (3) a physics-based Kramers–Kronig consistency constraint that regularizes frequency-dependent attenuation and delay. We evaluate our method over real-world data collected in multiple car cabins. Our approach significantly outperforms time- and hybrid-domain baselines on both simulated and real-world automotive datasets, cutting average magnitude and phase reconstruction errors by over 39% and 51%, respectively. Our experiments show that INFER achieves state-of-the-art performance frequency response modeling in automotive spaces.

## 1 Introduction

Accurate modeling of acoustic environments is fundamental to diverse applications, including architectural design, immersive audio rendering, and human–computer interaction Koyama et al. (2025). While techniques for room acoustics and open-field settings are relatively mature, car cabins have recently emerged as a critical yet underexplored application space. Unlike conventional rooms, cabins are compact, irregularly shaped enclosures with a heterogeneous mix of reflective and absorptive materials, whose acoustic responses are further complicated by their highly dynamic usage—seats recline, windows open, passengers Yoshimura et al. (2012). These properties create transfer characteristics that are difficult to capture using traditional measurement or simulation pipelines. At the same time, the acoustic environment inside the cabin is becoming central to the in-vehicle experience, enabling high-fidelity entertainment and safety-critical spatial alerts. Modern audio pipelines, such as Dolby Atmos and Sony 360 Reality Audio, aspire to deliver immersive sound in vehicles but require precise characterization of these transfer functions. Existing approaches rely on labor-intensive manual tuning, extensive in-situ measurements, or costly simulations based on idealized CAD models, all of which degrade under real-world perturbations. These factors collectively motivate the need for a data-driven, physically grounded, and adaptive modeling framework that can generalize across diverse cabin conditions while preserving perceptual fidelity and spatial audio quality.

Recent advances in neural implicit representations (INRs) Molaei et al. (2023); Zhang et al. (2025) provide a compelling alternative to handcrafted acoustic models. INRs learn continuous, resolution-agnostic mappings from spatial coordinates to signal values using multilayer perceptrons, enabling geometry-free reconstruction of complex fields. Extensions such as Neural Acoustic Fields (NAF) Luo et al. (2022), INRAS Su et al. (2022), and AV-NeRF Liang et al. (2023) have shown

that impulse responses can be compactly encoded by learning emitter–receiver transfer functions directly from data. Yet these models predominantly operate in the time domain and treat individual frequency components uniformly. For car cabins, however, both physical acoustics and human perception demand frequency-selective modeling: low-frequency modes dominate room-scale resonances, speaker crossovers introduce mid-band artifacts, and perceptual salience varies with spectral weighting of phase and magnitude.

Learning acoustic frequency response fields directly in the frequency domain can lead to fine-grained and physically-aware modeling of confined, resonant spaces such as car cabins. While frequency-domain representations have been explored for specific audio tasks (Lee & Lee, 2023; Di Carlo et al., 2024), existing neural acoustic field methods predict time-domain impulse responses and do not model continuous frequency response fields with spatial conditioning. By predicting each frequency bin independently, our approach naturally captures sharp spectral features and modal resonances that are often blurred in time-domain formulations. The spectral formulation also facilitates hardware-aware supervision: unreliable or unstable frequency bands can be identified and downweighted, and perceptually important regions—such as phase-sensitive low frequencies Oxenham (2018)—can be emphasized using auditory-inspired weighting schemes. Moreover, frequency-domain forward modeling encodes propagation delays exactly as phase shifts, eliminating the interpolation artifacts and discretization errors that plague time-domain approaches.

We propose **INFER** (**I**mplicit **N**eural **F**requency **R**esponse fields), a neural implicit representation framework that learns continuous frequency response fields in confined environments through fully spectral modeling. Our formulation couples a differentiable frequency-domain renderer with a complex-valued neural network that predicts the frequency response field conditioned on emitter location, receiver location, and their direction. We propose a novel frequency-specific spectral weighting for both phase and amplitude, enabling perceptual and hardware-aware loss design. Finally, we introduce physically grounded consistency constraints derived from Kramers–Kronig relations to regularize the joint behavior of attenuation and phase delay, leading to more interpretable and physically plausible reconstructions. Our approach delivers consistent performance boosts across the spectrum, outperforming the closest baseline by 39% in magnitude error and 51% in phase error on average (Table 4). These gains reflect both spectral fidelity and directional accuracy, and are visually evident in Fig. 1.

Our key contributions in this paper can be summarized as:

- A novel end-to-end frequency-domain modeling framework that parameterizes continuous frequency response fields with frequency-dependent complex attenuation in 3D space.

- A perceptually motivated, frequency-weighted supervision strategy that emphasizes critical bands and accounts for hardware artifacts such as crossovers and directivity lobes.

- A physically consistent formulation using Kramers–Kronig relations that jointly regularizes spectral amplitude and phase, enhancing generalization and interpretability.

- A comprehensive evaluation on both simulated and real car cabin datasets, demonstrating state-of-the-art phase and magnitude field reconstruction.

## 2 RELATED WORK

**Neural Implicit Representations for Physical Fields.** Neural implicit representations (INRs) have emerged as a powerful framework for modeling continuous physical signals by learning coordinate-to-signal mappings using multilayer perceptrons. Foundational methods such as SIREN Sitzmann et al. (2020) and Fourier feature encodings Tancik et al. (2020) allow compact modeling of high-frequency functions, enabling applications in 3D vision Mildenhall et al. (2021); Martel et al. (2021) and implicit surface reconstruction Wang et al. (2021). These ideas have been extended to domains like fluids Holl et al. (2020), mmWave Takawale & Roy (2025), and wave propagation Orekondy et al. (2023), demonstrating the versatility of coordinate-based learning. Our work builds on these insights and targets learning frequency response fields of spatially varying acoustic transfer functions within a confined car cabin, where modal resonances, material absorption, and directionality jointly shape the acoustic field. This setting requires jointly modeling amplitude and phase attenuation over a frequency spectrum.

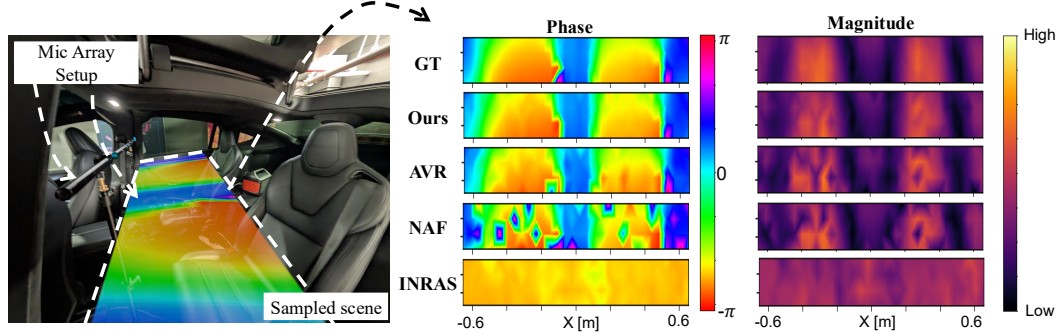

Figure 1: **Acoustic field modeling inside a car cabin. Left:** Measurement setup in the backseat of a Tesla Model X, where a speaker emits sound and spatial responses are recorded over a 2D grid. **Middle:** Spatial distribution of phase at 720Hz for ground truth (GT), our method (Ours), and baselines (AVR, NAF, INRAS). **Right:** Corresponding log-magnitude (energy) plots. Our method reconstructs smoother and physically consistent fields that preserve wavefront geometry and acoustic shadowing, outperforming baselines that exhibit artifacts or spatial inconsistency.

**Neural Modeling of Acoustic Fields.** Accurate modeling of sound fields in complex environments has traditionally relied on geometric acoustics, ray tracing, or numerical methods such as FEM and BEM Feng et al. (2013); Mo et al. (2015); Etgen & O'Brien (2007). These methods are physically grounded but computationally expensive and require detailed knowledge of geometry and material parameters. To address this, recent works like NAF Luo et al. (2022) and INRAS Su et al. (2022) have proposed neural implicit models to learn audio impulse responses from spatial data, often in time domain and with limited physical constraints. Recent methods have explored hybrid pipelines that uses both time-domain and frequency-domain forward modeling. For instance, AVR Lan et al. (2024) uses both sample delay and phase correction to model time-of-flight. However, AVR ultimately predicts time-domain impulse responses and assumes frequency-independent attenuation, limiting its ability to capture spectral variability across the scene. Furthermore, AVR employs uniform weighting across all frequencies during supervision, ignoring known variations in human perception and hardware characteristics. In contrast, our method INFER directly predicts frequency response fields, allowing frequency-aware supervision, perceptual spectral weighting, and physically grounded modeling of dispersion and absorption using Kramers–Kronig constraints—capabilities that existing impulse-response field prediction methods do not offer.

**Car Cabin Acoustic Modeling and Applications.** Acoustic field modeling inside car cabins presents unique challenges due to the confined space, material heterogeneity, complex modal behavior, and intricate reflection patterns Accardo et al. (2018); Yoshimura et al. (2012). Classical simulation techniques based on FEM or BEM Wang et al. (2013); Liu et al. (2017) are accurate but computationally prohibitive for design iteration or personalization. Empirical IR measurements Farina et al. (1998) and equalization techniques often ignore the global structure of the acoustic field, focusing instead on specific locations. Recent learning-based methods lack frequency-aware modeling and typically neglect physically grounded constraints essential for accurate modeling. Our method addresses these limitations by learning a continuous, physically grounded frequency-domain representation of the cabin's 3D acoustic field, enabling accurate reconstruction of both amplitude and phase, with explicit modeling of dispersion, crossover behavior, and material absorption—critical features not captured by prior empirical or neural approaches.

## 3 PRIMER: PHYSICS OF ACOUSTIC PROPAGATION IN LOSSY MEDIA

### 3.1 PROBLEM PREMISE

Achieving physically consistent and perceptually accurate acoustic modeling in confined spaces like car cabins requires a deeper understanding of how sound propagates in complex, lossy media. Unlike free-field environments, car interiors exhibit complex modal behavior, intricate multipath interference, and frequency-dependent absorption—making frequency-domain analysis not just convenient but essential. As established in Sec. 1, our method adopts a frequency-by-frequency modeling ap-

proach that gives the neural network the flexibility to understand these phenomena. To motivate and ground our spectral formulation, this section introduces the three key physical concepts underpinning our design: (1) how free-field propagation naturally maps to a frequency-domain formulation, (2) how rich multipath effects can be modeled via the Huygens–Fresnel principle, and (3) how signals interact in real media and experience attenuation and dispersion.

### 3.2 FREE-FIELD PROPAGATION AND ITS FREQUENCY DOMAIN REPRESENTATION

To understand acoustic propagation, we begin with the classical time-domain free-space model. When a point source at $\mathbf{p}_{\text{tx}} \in \mathbb{R}^3$ emits an impulse at $t = 0$, the pressure at a receiver located at $\mathbf{p} \in \mathbb{R}^3$ in an ideal, lossless medium experiences decay in energy and arrives with a delay and is given by the 3D Green's function Kuttruff (2016):

$$h(t) = \frac{1}{4\pi r}\delta\left(t - \frac{r}{v}\right), \quad r = \|\mathbf{p} - \mathbf{p}_{\text{tx}}\|, \quad v = \text{speed of sound} \tag{1}$$

The energy decay is due to $1/r$ spherical spreading of the pressure wave and the delay corresponds to the time-of-flight $r/v$. To arrive at the frequency domain representation of this phenomena, we apply the Fourier transform which yields : $H(f) = \frac{1}{4\pi r}\exp\left(-j\frac{2\pi f r}{v}\right)$, The magnitude remains governed by $1/r$, while the propagation delay is now expressed as a frequency specific phase shift $e^{-j\omega\tau}$, where $\omega = 2\pi f$. This forms the building block of our frequency domain rendering approach.

### 3.3 THE HUYGENS–FRESNEL PRINCIPLE FOR MULTIPATH EFFECT MODELING

Acoustic wavefields in confined spaces arise from intricate multi-path interactions involving reflections, diffractions, and scattering. To capture this behavior, we draw inspiration from the Huygens–Fresnel principle Lian (2023), which posits that each point on a wavefront acts as a secondary emitter. In the frequency domain, the resulting complex pressure at a point $\mathbf{x}$ can be modeled as:

$$P(\mathbf{x}, \omega) = \int_\Omega G(\mathbf{x}, \mathbf{x}'), S(\mathbf{x}', \omega), d\mathbf{x}', \tag{2}$$

where $G(\mathbf{x}, \mathbf{x}')$ is the Green's function encoding phase and amplitude propagation from $\mathbf{x}'$ to $\mathbf{x}$, $\Omega \in \mathbb{R}^3$ is the volume being modelled, and $S(\mathbf{x}', \omega)$ is the frequency-domain strength of secondary emission. In practice, for lossy media, $G(\mathbf{x}, \mathbf{x}')$ cannot be computed analytically and depends on the spatial distribution of material properties along the propagation path. In Section 3.4, we introduce the local complex attenuation field $\delta(f, x)$, whose path-integrated effects determine the effective Green's function between any two points. This formulation motivates our design: instead of tracing discrete reflection paths, we model the volume as a continuous field of directional secondary emitters. Each voxel learns to re-radiate incoming energy in all directions, capturing reverberation and scattering in a physically grounded, data-driven manner.

### 3.4 ATTENUATION AND DISPERSION IN MEDIA

Real acoustic environments are inherently lossy. As the sound propagates, amplitudes decay due to absorption and scattering (*attenuation*) and phases evolve at frequency-dependent speeds (*dispersion*). Crucially, these two effects are not independent artifacts - attenuation and dispersion are *inherently linked*. In any linear, time-invariant medium, the way amplitude varies with frequency determines how phase varies with frequency (and vice versa); one cannot be chosen independently of the other. This coupling is formalized by the Kramers–Kronig (KK) relation O'Donnell et al. (1981). Practically, this matters because past models fit only amplitude decay which, by construction, miss the paired frequency-dependent phase response that a real medium must exhibit.

**Kramers–Kronig relations.** The Kramers–Kronig relations express that the phase-bearing and amplitude-bearing parts of the medium's correction to wavenumber as Hilbert-transform. Physically, they ensure that no component of the response can occur before its excitation, i.e., *causality*. In acoustics, frequency-dependent propagation is written via a complex wavenumber $k(\omega) = k_0(\omega) + \delta(\omega)$, $k_0(\omega) = \omega/v$, where $\delta(\omega) = \text{Re}\,[\delta(\omega)] + j\,\text{Im}\,[\delta(\omega)]$ captures medium-induced modifications. The KK relations impose

$$\text{Re}\,[\delta(\omega)] = \frac{1}{\pi}\mathcal{P}\int_{-\infty}^{\infty}\frac{\text{Im}\,[\delta(\omega')]}{\omega' - \omega}\,d\omega', \quad \text{Im}\,[\delta(\omega)] = -\frac{1}{\pi}\mathcal{P}\int_{-\infty}^{\infty}\frac{\text{Re}\,[\delta(\omega')]}{\omega' - \omega}\,d\omega'. \tag{3}$$

**Complex attenuation fields.** We operationalize this principle by predicting, at each spatial point, a *complex attenuation* that separates amplitude loss and phase modification: $\delta(f, \mathbf{x}) = \sigma(f, \mathbf{x}) + j\,\beta(f, \mathbf{x})$, where $\sigma \geq 0$ is the absorption coefficient and $\beta$ encodes dispersion-induced phase-velocity deviation. The KK-consistency is maintained through the Kramers–Kronig consistency regularizer explained in 4.3.

**Physically consistent volume rendering.** Once $\delta$ is known locally, its effects *accumulate* along a path as multiplicative transmittance and additive phase. For a small segment of length $\Delta u$,

$$T_{\text{mat}}(\Delta u) = \exp(-\delta\,\Delta u) = \underbrace{\exp(-\sigma\,\Delta u)}_{\text{amplitude decay}} \cdot \underbrace{\exp(j\,\beta\,\Delta u)}_{\text{phase shift}}. \tag{4}$$

Over a full path $\mathbf{p}(s)$ of length $L$, amplitude and phase accumulate as $T_{\text{amp}} = \exp\!\left(-\int_0^L \sigma\big(f, \mathbf{p}(s)\big)\,ds\right)$, and $\phi_{\text{mat}} = -\int_0^L \beta\big(f, \mathbf{p}(s)\big)\,ds$. Prior acoustic neural fields typically model absorption or overall amplitude but ignore the causally paired, frequency-dependent phase response. In contrast, we are the first to encode KK-consistent complex attenuation in a neural acoustic renderer, preventing non-physical phase behavior and improving both interpretability and generalization.

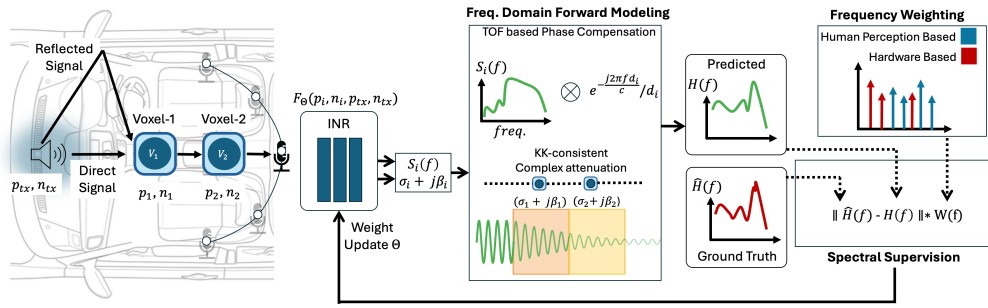

Figure 2: **System Overview.** Illustration of our frequency-domain acoustic forward model. For each point sampled along rays cast from the microphone, the MLP predicts a frequency-domain signal and attenuation. A TOF: time-of-flight based phase shift is applied to the signal and material based absorption and phase shifts are applied to produce the final response by accumulating signal from all directions.

## 4 METHODS FOR IMPLICIT NEURAL FREQUENCY RESPONSE FIELD

We introduce INFER, a fully frequency-domain neural rendering framework for modeling spatially varying frequency response fields in real-world environments. Unlike prior time-domain methods, INFER directly learns complex frequency responses—capturing sub-sample propagation delays, frequency-dependent attenuation, and dispersive phase shifts. This allows flexible perceptual weighting across frequencies, such as down-weighting hardware-specific crossover bands or emphasizing phase at low frequencies for localization. Fig. 2 gives a brief overview of INFER.

### 4.1 NEURAL FIELD PARAMETERIZATION

Given a scene with a sound source located at $\mathbf{p}_{\text{tx}}$ and oriented along the unit vector $\hat{\mathbf{n}}_{\text{tx}}$, we define a neural field $F_\Theta$ that predicts the local frequency-domain behavior at any spatial query point $\mathbf{p}$ and frequency $f$:

$$F_\Theta : (\mathbf{p}, \hat{\mathbf{n}}, \mathbf{p}_{\text{tx}}, \hat{\mathbf{n}}_{\text{tx}}) \mapsto \{\delta(f, \mathbf{p}),\ S(f, \mathbf{p}, \hat{\mathbf{n}})\} \tag{5}$$

Here, $\delta(f, \mathbf{p}) \in \mathbb{C}$ is the complex attenuation encoding the frequency-dependent transmission loss at point $\mathbf{p}$, and $S(f, \mathbf{p}, \hat{\mathbf{n}}) \in \mathbb{C}$ is the directional spectrum re-radiated from that point toward the unit vector $\hat{\mathbf{n}}$. Together, they fully characterize how an incoming acoustic wave is transformed and retransmitted from each location in the volume. Unlike other neural acoustic rendering models, INFER predicts all quantities directly in the frequency domain. The goal of the neural field is thus to answer: given a source at $(\mathbf{p}_{\text{tx}}, \hat{\mathbf{n}}_{\text{tx}})$, what frequency domain signal is re-emitted in any direction

$\hat{\mathbf{n}}$ from point $\mathbf{p}$, and what is the frequency-specific material-induced attenuation along the way? We implement $F_\Theta$ using a two-branch architecture. The first branch takes $(\mathbf{p}, \mathbf{p}_{\text{tx}})$ and predicts $\delta(f, \mathbf{p})$ via a material sub-network. The second branch conditions on the learned features from $\delta$, the receiver direction $\hat{\mathbf{n}}$, and source direction $\hat{\mathbf{n}}_{\text{tx}}$, and predicts the directional retransmission spectrum $S(f, \mathbf{p}, \hat{\mathbf{n}})$. This decomposition reflects the physical structure of the problem: attenuation is direction-independent, while retransmission is highly directional.

## 4.2 FREQUENCY-DOMAIN RENDERING

Our goal is to predict frequency response at any receiver location. To synthesize the acoustic frequency response at a receiver location $\mathbf{p}_{\text{rx}}$, we cast rays in direction $\hat{\mathbf{n}}$ and sample $N$ points along the ray as $\mathbf{p}_k = \mathbf{p}_{\text{rx}} + u_k\hat{\mathbf{n}}$. We discuss the ray marching strategy in detail in Sec A.4 of the Appendix. At each sampled point, we query the neural field to evaluate local frequency-domain properties and accumulate their contributions using a physically motivated rendering equation:

$$H_{\hat{\mathbf{n}}}(f) = \sum_{k=1}^{N} S_k(f) \cdot \frac{1}{4\pi u_k} \cdot e^{-j2\pi f u_k/v} \cdot e^{j\phi_k(f)} \cdot \alpha_k T_k \tag{6}$$

This equation models how sound emitted from the transmitter propagates through the environment and contributes to the received frequency response along direction $\hat{\mathbf{n}}$. At each sampled point, $S_k(f)$ denotes the local directional spectrum predicted by the neural field, $e^{-j2\pi f u_k/v}$ introduces the phase shift due to time-of-flight delay, and $\frac{1}{u_k}$ accounts for spherical spreading via distance-based amplitude decay. The term $\alpha_k = 1 - \exp(-\sigma_k \Delta u_k)$ represents the discrete opacity arising from local absorption, while $T_k = \prod_{j<k}(1 - \alpha_j)$ captures accumulated transmittance from earlier samples along the ray. Finally, $\phi_k(f) = \sum_{j<k} \beta_j \Delta u_j$ models the cumulative phase shift induced by dispersive propagation through the medium. Together, these terms account for direction-dependent emission, distance-based decay, frequency-selective absorption, and phase dispersion—without discretizing time or relying on post-hoc transforms. To model a realistic microphone, which integrates sound from multiple directions, we perform weighted integration over a discrete set of directions: $H(f) = \sum_m G(\hat{\mathbf{n}}_m) H_{\hat{\mathbf{n}}_m}(f)$, where $G(\hat{\mathbf{n}}_m)$ models microphone directivity.

## 4.3 SPECTRAL SUPERVISION

A central design choice in our framework lies in how we supervise the learning of complex acoustic responses in the frequency domain. Rather than ultimately predicting time-domain impulse responses and deriving frequency behavior indirectly—as in prior works—we operate entirely in the spectral domain and define a suite of loss functions that target perceptual alignment, hardware-aware weighting, and physically consistent attenuation. Let $H(f)$, $\hat{H}(f) \in \mathbb{C}$ denote the ground-truth and predicted complex frequency responses at a receiver, and let $w(f) \geq 0$ denote a frequency-dependant weight that can encode hardware or perceptual importance.

**Weighted complex, magnitude, and phase losses.** We decompose the spectral supervision into three complementary terms: one for the real and imaginary parts (denoted by $Re[.]$ and $Im[.]$), one for the magnitude, and one for phase. These jointly ensure accurate complex-valued reconstruction while allowing flexible emphasis through $W_{spec}(f)$, $W_{mag}(f)$ and $W_{phase}(f)$:

$$L_{\text{spec}} = \sum_f W_{spec}(f) \left( |\text{Re}[H(f)] - \text{Re}[\hat{H}(f)]| + |\text{Im}[H(f)] - \text{Im}[\hat{H}(f)]| \right), \tag{7}$$

$$L_{\text{mag}} = \sum_f W_{mag}(f) \left| |H(f)| - |\hat{H}(f)| \right|, \tag{8}$$

$$L_{\text{phase}} = \sum_f W_{phase}(f) \left( |\cos \angle H(f) - \cos \angle \hat{H}(f)| + |\sin \angle H(f) - \sin \angle \hat{H}(f)| \right). \tag{9}$$

These losses provide fine-grained frequency-level control. For example, frequencies in crossover regions of a speaker can be downweighted to avoid unstable learning, while perceptually important midbands can be emphasized.

**Spectral envelope smoothing.** Acoustic spectra in real-world environments often exhibit narrow-band fluctuations due to interference, which are perceptually less important than the broader spectral shape. Inspired by standard practices in audio engineering and car-cabin equalization, we regularize the predicted and ground-truth log-magnitude spectra while calculating envelope loss $L_{env}$ using an exponential smoothing filter $\mathcal{E}$:

$$L_{\text{env}} = \sum_f \left| \mathcal{E}\left(\log(|H(f)| \cdot w(f) + \epsilon)\right) - \mathcal{E}\left(\log(|\hat{H}(f)| \cdot w(f) + \epsilon)\right) \right|, \tag{10}$$

where $\epsilon$ is a small positive constant to avoid singularities. This regularization encourages fidelity to the broadband spectral envelope while tolerating harmless fine-grained ripples, leading to smoother convergence and perceptually cleaner reconstructions.

**Kramers–Kronig consistency regularizer.** As introduced in Sec. 3.4, in physical media, frequency-dependent attenuation and dispersion are coupled by the Kramers–Kronig (KK) relations. To enforce this constraint in learning, we define:

$$\hat{\beta}(f) = \mathcal{H}\{\sigma\}(f), \qquad L_{\text{KK}} = \sum_{f \in \mathcal{B}} \left(\beta(f) - \kappa\,\hat{\beta}(f)\right)^2, \tag{11}$$

where $\mathcal{H}$ is a discrete Hilbert transform implemented via two-sided even extension and a raised-cosine taper, $\kappa$ is a learnable scalar for scaling alignment, and $\mathcal{B}$ is a frequency band mask to exclude unreliable bins (e.g., DC/Nyquist). This term ensures the learned attenuation field respects causality and avoids unphysical phase artifacts.

**Total objective.** The complete spectral loss is a weighted combination of the above components:

$$L_{\text{total}} = \lambda_{\text{spec}} L_{\text{spec}} + \lambda_{\text{mag}} L_{\text{mag}} + \lambda_{\text{phase}} L_{\text{phase}} + \lambda_{\text{env}} L_{\text{env}} + \lambda_{\text{KK}} L_{\text{KK}} + L_{\text{aux}}, \tag{12}$$

where $\lambda_{\{\cdot\}}$ are hyperparameters controlling the contribution of each term. $L_{\text{aux}}$ denotes auxiliary loss terms. $L_{\text{aux}}$ is carried over from prior work Yamamoto et al. (2020); Majumder et al. (2022) and consists of multi-resolution STFT loss and energy-shape penalties, and are used for stability rather than driving the primary supervision. Together, these losses constitute a principled and physically grounded spectral supervision strategy. They allow our model to align with both perceptual and physical constraints—capturing sharp resonances, respecting causal propagation, and adapting to hardware-specific frequency responses—while operating entirely in the frequency domain.

## 5 EXPERIMENTS

We evaluate INFER on the task of reconstructing the 3D frequency response field inside car cabins. Given measured impulse responses at 48 kHz, the model is trained to predict the complex frequency-domain response at unseen receiver positions. We compare INFER to prior state-of-the-art methods—NAF, INRAS, and AVR—focusing on reconstruction accuracy across frequency bands.

### 5.1 DATASETS

**Simulated.** We evaluate our method on both simulated and real-world datasets. The simulated data is generated using COMSOL's *Car Cabin Acoustics—Transient Analysis* module, which solves the time-dependent wave equation with realistic, frequency-dependent boundary admittances. We extract impulse responses at 216 receiver positions across the cabin geometry.

**Real.** For real-world evaluation, we collect measurements in both the BUCK vehicle mock-up and a Tesla Model X using five loudspeakers and a 16-channel UMA-16 microphone array. We record 4096-sample IRs at 48 kHz with physically measured speaker and microphone positions. Fig. 3 shows the data collection environment and hardware for both BUCK(left) and Tesla model X(right).

### 5.2 IMPLEMENTATION DETAILS

The input to our model consists of a 3D query point $\mathbf{p} \in \mathbb{R}^3$, transmitter location $\mathbf{p}_{\text{tx}} \in \mathbb{R}^3$, and directions $(\hat{\mathbf{n}}_{\text{tx}}, \hat{\mathbf{n}}) \in \mathbb{R}^3$ representing the emitter and receiver orientations. All input coordinates are encoded using a hash grid based encoding. The model outputs the corresponding complex attenuation $\delta[f] \in \mathbb{C}^{\mathbb{T}}$ and directional spectrum $S[f] \in \mathbb{C}^{\mathbb{T}}$ at that query point, from which the frequency

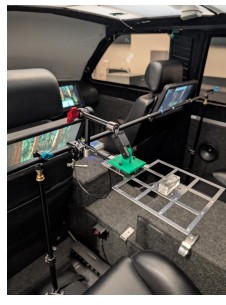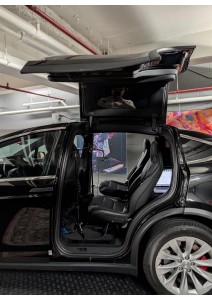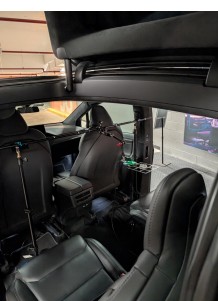

Figure 3: **Data collection setup.** (a)Left: Data is collected in controlled environment - The BUCK, which is a vehicle mockup with realistic car interior and acoustic frontend. (b)Right: Data is also collected in real environment - Tesla Model X.

Table 1: Mean Absolute Error for Metrics for BUCK and Tesla setups (lower is better).

| Method | Buck | | | | | | | | Tesla | | | | | | | |
|---|---|---|---|---|---|---|---|---|---|---|---|---|---|---|---|---|
| | Amp | Ang | Spec | STFT | Ene. | Env. | T60 | EDT | Amp | Ang | Spec | STFT | Ene. | Env. | T60 | EDT |
| INRAS | 0.29 | 1.54 | 0.9 | 1.6 | 7.13 | 2.79 | 3.0 | 3.0 | 0.43 | 1.63 | 1.0 | 2.2 | 3.93 | 3.82 | 14.6 | 109.8 |
| NAF | 0.14 | 0.54 | 0.3 | 1.0 | **5.55** | 1.13 | **1.3** | **1.7** | 0.48 | 1.63 | 1.2 | 2.2 | 2.25 | 4.13 | 10.0 | 8.1 |
| AVR | 0.21 | 0.81 | 0.5 | 1.5 | 7.95 | 2.06 | 3.2 | 2.4 | 0.28 | 1.61 | 1.0 | 2.7 | 5.28 | 6.92 | 49.6 | 24.0 |
| Ours | **0.12** | **0.50** | **0.2** | 1.2 | 5.56 | **0.95** | 9.8 | 2.6 | **0.14** | **0.59** | **0.3** | **1.0** | **1.57** | **1.45** | **8.4** | **4.0** |

response $H[f]$ is rendered using Eq.6. The model is implemented as MLPs with 6 fully connected layers and 256 hidden units per layer, using ReLU activations. Rendering is performed using ray marching with 64 points per ray, accumulating complex-valued attenuation and delay across the path as described in Sec. 4.2. We integrate over $64 \times 32$ azimuth–elevation rays per receiver to form the output signal. We train all models using the Adam optimizer with an initial learning rate of $5 \times 10^{-4}$. Training takes approximately 24 hours on a single NVIDIA L40 GPU. All baseline models are trained with the same network size and data splits for fair comparison.

## 5.3 QUANTITATIVE RESULTS

Across both BUCK and Tesla, INFER achieves the lowest errors on the core frequency-domain metrics (amplitude, angle, spectral, envelope), outperforming NAF, INRAS, and AVR (Table 1). On BUCK, our model is competitive on energy but lags on time-domain reverberation metrics (T60, EDT), which is consistent with our frequency-centric supervision. On Tesla, INFER also leads on the time-domain metrics (lowest T60 and EDT), indicating better generalization. Per-frequency analysis (Table 4) shows consistent gains: INFER attains the best magnitude and phase errors at every reported band, with particularly large phase advantages at low frequencies (e.g., 180 Hz: 0.029 vs. 0.076 for the next best), while maintaining the best average magnitude error. The details on metric calculation can be found in Appendix 8.

**Generalizibility.** Although, INFER targets modeling frequency responses in closed confines spaces like car cabins as its target application, the underlying technique has no components specific to car cabin modeling. Thus, to prove the generalizibility of our technique, we evaluate it on room scale datasets such as MeshRIR Koyama et al. (2021), RAF-Furnished, and RAF-Empty Chen et al. (2024) as shown in Table 2. INFER achieves the best or tied-best performance across these settings. We also show that incorporating KK consistency regularizer improves performance compared to and the KK-free ablation.

**Ablation Study.** We ablate over different scene sampling strategies, training data sparsity and loss components. These results can be found in Table 3.

- **Loss Components**: We find that removing any term degrades performance across all metrics (with only minor fluctuations in $T_{60}$), indicating that the full multi-objective loss is

essential for accurately constraining both magnitude and phase behavior, while $T_{60}$ is less sensitive and can slightly improve when the network underfits specific frequency bands.

- **Dataset Sparsity**: We ablate training-data sparsity and observe a smooth degradation in performance as the data become increasingly sparse, indicating that INFER degrades gracefully under reduced sampling density.

- **Sampling parameters**: We vary the number of rays (both azimuth and elevation) and points per ray. We find that both increasing the ray numbers and the sampling points will both enhance performance, but come with the cost of low training speed and high memory consumption

Table 2: Evaluation on room-scale environments.

| Method | MeshRIR | | | | | | RAF-Furnished | | | | | | RAF-Empty | | | | | |
|---|---|---|---|---|---|---|---|---|---|---|---|---|---|---|---|---|---|---|
| | Phase | Amp. | Env. | T60 | C50 | EDT | Phase | Amp. | Env. | T60 | C50 | EDT | Phase | Amp. | Env. | T60 | C50 | EDT |
| AAC-nearest | 1.47 | 0.91 | 1.40 | 8.6 | 2.20 | 58.8 | 1.60 | 1.09 | 4.83 | 13.0 | 3.41 | 73.5 | 1.60 | 1.09 | 4.83 | 13.0 | 3.41 | 73.3 |
| AAC-linear | 1.44 | 0.89 | 1.42 | 8.2 | 2.29 | 58.9 | 1.60 | 0.99 | **3.81** | 12.4 | 3.65 | 90.2 | 1.59 | 1.10 | 5.22 | 13.1 | 3.25 | 71.5 |
| Opus-nearest | 1.45 | 0.72 | 1.37 | 5.2 | 1.26 | 35.7 | 1.60 | 1.19 | 5.35 | 14.4 | 3.78 | 80.3 | 1.59 | 1.16 | 4.58 | 13.3 | 4.25 | 100.6 |
| Opus-linear | 1.43 | 0.69 | 1.37 | 6.9 | 1.83 | 49.3 | 1.60 | 1.47 | 5.74 | 13.1 | 3.55 | 77.8 | 1.59 | 0.95 | **4.26** | 12.7 | 3.94 | 95.5 |
| NAF | 1.61 | 0.64 | 1.59 | 4.2 | 1.25 | 39.0 | 1.62 | 0.93 | 5.34 | 7.1 | **0.98** | **20.6** | 1.62 | 0.85 | 4.67 | 8.0 | 1.22 | 26.3 |
| INRAS | 1.61 | 0.77 | 1.85 | 3.4 | 1.47 | 40.7 | 1.62 | 0.96 | 6.43 | 6.9 | 1.08 | 21.4 | 1.62 | 0.88 | 4.72 | 7.6 | 1.21 | 25.8 |
| AVR | 1.48 | 0.54 | **1.15** | 3.9 | 0.92 | 35.1 | **1.58** | 0.28 | 5.79 | 6.6 | 1.12 | 22.88 | **1.58** | 0.29 | 5.16 | 6.3 | 1.18 | 24.3 |
| INFER (without KK) | 1.224 | 0.26 | 7.40 | **2.8** | 0.57 | 12.71 | **1.58** | 0.2337 | 5.33 | 6.35 | 1.15 | 22.56 | **1.58** | 0.23 | 4.73 | **6.0** | 1.07 | **23.1** |
| INFER | **1.194** | **0.24** | 7.34 | 3.14 | **0.50** | **12.45** | **1.58** | **0.2197** | 5.40 | **6.34** | 1.08 | 22.17 | **1.58** | **0.23** | 4.76 | 6.3 | 1.11 | 23.5 |

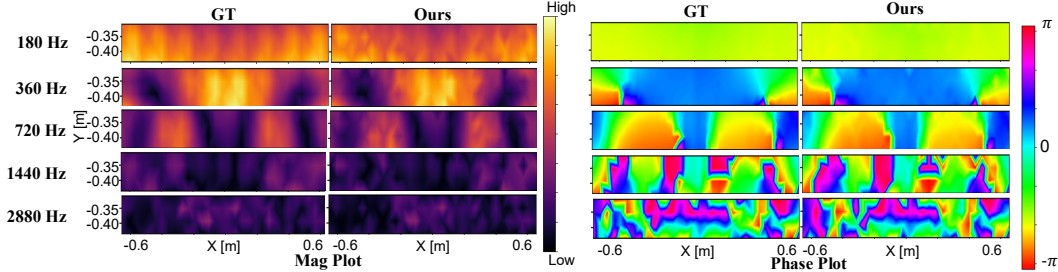

Figure 4: **Qualitative results.** (Left) Spatial plots comparing ground truth (GT), and our method's ability to reconstruct magnitude and phase field across various frequency bands.

## 5.4 QUALITATIVE RESULTS

**Baseline comparison (Fig. 1).** At 720 Hz the cabin exhibits mid–high modal density with pronounced interference and wrapped phase discontinuities. The left panel (magnitude) shows two dominant high–energy lobes separated by low–energy troughs in the ground truth. Our method reproduces both the location and contrast of these structures, preserving the nodal troughs and avoiding spurious speckle. AVR and NAF show over–textured patterns with high–frequency artifacts and INRAS collapses toward a nearly uniform field. The right panel (phase) smoothly varying phase across two regions. Our prediction aligns with both in terms of the global gradient and the placement, yielding continuous phase evolution. AVR and NAF introduce local phase jitter and INRAS is nearly constant, indicating it does not recover the true phase evolution at this frequency.

**Across-frequency behavior (Figs. 4).** We evaluate a wide range of frequencies that transition from low–modal regimes to highly complex interference patterns. At 180 Hz the ground truth is dominated by a slowly varying field; our model reproduces this near-uniform phase and the smooth, weakly varying magnitude without introducing spurious structure. At 360–720 Hz, where the cabin begins to exhibit distinct standing-wave patterns, our reconstructions recover both the placement and contrast of high/low-energy lobes together with the associated phase gradients and wrap seams. At higher frequencies (1440–2880 Hz) the field contains rapid spatial oscillations and multiple discontinuities. Despite this, our predictions remain stable: magnitude maps retain fine-scale contrast without speckle or over-smoothing, and phase maps capture the correct number and placement of wraps with coherent local gradients.

Table 3: Model ablations. Performance for the model variants on Buck dataset.

| Study Objectives | Variation | Phase. | Amp. | Env. | T60 | EDT |
|---|---|---|---|---|---|---|
| Loss Component | w/o mag loss | 0.74 | 0.18 | 1.6 | 8.9 | 7.1 |
| | w/o phase loss | 0.98 | 0.2 | 1.8 | 7.8 | 6.2 |
| | w/o energy loss | 0.5 | **0.12** | 0.99 | 24.0 | 3.5 |
| | w/o kk loss | 0.77 | 0.18 | 1.7 | 9.8 | 7.3 |
| | w/o stft loss | 0.64 | 0.15 | 1.4 | 7.0 | 4.4 |
| | w/o spec loss | 1.44 | 0.25 | 2.6 | **2.4** | 2.7 |
| | w/o env loss | 0.57 | 0.13 | 1.2 | 5.7 | 3.9 |
| | w/o frequency weighting | 0.55 | 0.13 | 1.2 | 2.8 | 3.4 |
| | w/ all loss components | **0.48** | **0.12** | **0.95** | 9.8 | **2.6** |
| Training Data Reduction | 30% | 1.08 | 0.25 | 1.9 | 3.6 | 4.4 |
| | 50% | 0.81 | 0.19 | 1.4 | **3.2** | 3.7 |
| | 60% | 0.68 | 0.15 | 1.3 | 3.5 | 3.7 |
| | 75% | **0.5** | **0.12** | **0.95** | 9.8 | **2.6** |
| Sampling Parameters | 32 × 16 rays, 64 points | 0.98 | 0.43 | 4.2 | 13.6 | 10.2 |
| | 48 × 24 rays, 64 points | 0.91 | 0.24 | 1.9 | 10.06 | 6.1 |
| | 64 × 32 rays, 64 points | 0.5 | 0.12 | 0.95 | 9.8 | **2.6** |
| | 64 × 32 rays, 40 points | 1.13 | 0.32 | 2.3 | **7.0** | 6.9 |
| | 64 × 32 rays, 70 points | **0.48** | **0.11** | **0.91** | 7.8 | 2.8 |

Table 4: Per-frequency Mean Absolute Error for Buck Dataset (lower is better).

| Method | Mag Err | | | | | | Phase Err | | | | | |
|---|---|---|---|---|---|---|---|---|---|---|---|---|
| | 180 | 360 | 720 | 1440 | 2880 | Avg | 180 | 360 | 720 | 1440 | 2880 | Avg |
| INRAS | 0.751 | 0.602 | 0.662 | 0.384 | 0.280 | 0.536 | 0.100 | 0.824 | 1.232 | 1.285 | 1.376 | 0.963 |
| NAF | 0.331 | 0.277 | 0.242 | **0.154** | 0.150 | 0.231 | 0.076 | 0.284 | 0.398 | 0.500 | 0.413 | 0.334 |
| AVR | 0.465 | 0.388 | 0.255 | 0.183 | 0.217 | 0.302 | 0.105 | 0.187 | 0.151 | 0.330 | 0.665 | 0.288 |
| Ours | **0.149** | **0.152** | **0.125** | 0.154 | **0.118** | **0.140** | **0.029** | **0.076** | **0.081** | **0.194** | **0.322** | **0.140** |

## 6 DISCUSSION AND FUTURE WORK

This work introduces INFER, a novel spectral-domain neural representation for car cabin acoustics, enabling accurate frequency response reconstruction from sparse measurements. Our method surpasses prior baselines in both magnitude and phase fidelity, and remains physically grounded through causality-aware regularization. Beyond its immediate impact on spatial audio modeling and personalization, our approach opens avenues for integrating learned acoustic fields into downstream tasks such as adaptive ANC, directional speech enhancement, and real-time audio rendering. Future extensions include joint modeling across passenger positions, integrating speaker-specific transfer functions, and exploring generalization to unseen vehicle geometries or dynamic cabin conditions.

## 7 CONCLUSION

We introduced INFER, a novel spectral-domain framework that models acoustic propagation in confined environments using implicit neural representations. By operating directly in the frequency domain, our method enables perceptually grounded supervision, hardware-aware weighting, and physically consistent regularization through the Kramers–Kronig constraint. Our differentiable renderer explicitly accounts for phase and attenuation via complex-valued ray integration, yielding spatially coherent and frequency-resolved reconstructions. Extensive evaluations on real and simulated car cabin datasets demonstrate that INFER substantially outperforms prior time-domain and hybrid approaches, achieving over 50% improvement in phase accuracy and 39% in magnitude fidelity relative to the best baseline. We believe this work paves the way for accurate and physically grounded neural acoustic modeling in automotive spaces.

## 8 REPRODUCIBILITY

Anonymized code and demo datasets will be available on our webpage (`https://anonymous1415510-spec.github.io`). We provide details about comparison with other algorithms to facilitate reproducing our results. All details about the hyperparameters, environment specifications, and real-world experiment setup are provided in the appendix or the website.

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

## DISCLOSURE OF LLM USAGE FOR WRITING

Large Language Models (LLMs) were used solely for grammar refinement and polishing of the manuscript text. All ideas, technical content, experimental design, and analysis were independently developed by the authors without LLM assistance.

## TRAINING DETAILS AND REPRODUCIBILITY

### A.1 REPRODUCIBILITY

Anonymized code and demo datasets will be available on our webpage (`https://anonymous1415510-spec.github.io`). We provide details about comparison with other algorithms to facilitate reproducing our results. All details about the hyperparameters, environment specifications, and real-world experiment setup are provided in the appendix or the website.

### A.2 MODEL ARCHITECTURE, DATASETS AND RENDERER DETAILS

We adopt the `AVRModel_complex_FD_FreqDep_PhaseCorrection` model architecture with the `AVRRenderFD_FreqDep_PhaseCorrection_KK` renderer. This setup is designed for complex-valued frequency domain rendering and enables physically grounded learning with explicit modeling of attenuation and phase velocity. Fig. 5 provides additional details on the input to each module of the network.

**Dataset details**: INFER has been trained on three datasets mentioned in . INFER is trained independently in each dataset. Each dataset is divided in 75:15:10 for training set, testing set and validation set. We provide details of spatial sampling in Table 6

**Key architectural components**:

- **Complex Frequency Field Prediction**: The model learns frequency-dependent attenuation fields $\delta(f) = \sigma(f) + j\beta(f)$ and complex responses $H(f)$ using MLPs operating on hash-encoded spatial coordinates.
- **Separate Signal and Attenuation Networks**: Frequency-specific signal and attenuation values are predicted using distinct encoders and MLPs.
- **Directional Encodings**: Transmitter and receiver directions are encoded using spherical harmonics.
- **Renderer Pipeline**: Rays are sampled in spherical directions with integration over 64 azimuth × 32 elevation rays, each with 64 samples from near=0 to far=4 meters. Cumulative attenuation is applied using $\exp(-\sum \sigma_i \Delta u_i + j \sum \beta_i \Delta u_i)$.
- **Model Size**: The Attenuation Networks contains 670,978 parameters (76,416 for encoder (3 layers 128 neurons) + 594,562 decoder (3 layers 128 neurons)). The Retransmission Network has 2,840,578 parameters ((3 layers 512 neurons)). Thus the total parameters are 3,511,556 parameters ( 3.51M).

| Parameter | Value |
|---|---|
| Learning Rate | $5 \times 10^{-4}$ (cosine annealing to $5 \times 10^{-5}$) |
| Optimizer | Adam |
| Total Iterations | 15,000 |
| Batch Size | 1 |
| Rendering Samples | 64 per ray |
| Azimuth × Elevation Rays | $64 \times 32$ |
| Speed of Sound | 343.8 m/s |
| Sampling Frequency | 48,000 Hz |
| Path Loss Exponent | 1 |
| Layers | 8 fully connected layers |
| Hidden Units | 256 neurons per layer |
| Activation | ReLU |
| Positional Encoding | 10 frequencies for spatial and directional input |

Table 5: Training hyperparameters and network architecture for INFER.

Table 6: Dataset Spatial Sampling Characteristics.

| Dataset | Mic locs | Speaker locs | Horizontal spacing | Vertical levels |
|---|---|---|---|---|
| COMSOL | 216 | 1 | 9.0 cm | 3 |
| Tesla | 384 | 5 | 4.2 cm | 3 |
| Buck | 384 | 5 | 4.2 cm | 3 |

## A.3 ADDITIONAL EVALUATIONS

We provide results to additional evaluations here -

- Spatial error plot: Figure 6 provides a spatial visualization of magnitude and phase reconstruction errors across the measurement plane for all baseline methods. INFER exhibits uniformly low error with minimal spatial structure, indicating that the learned frequency-response field interpolates smoothly across unobserved receiver positions. In contrast, competing methods show localized regions of high magnitude and phase error, reflecting their difficulty in modeling fine-grained spatial acoustic variation.

- Evaluation on COMSOL dataset: We additionally evaluate INFER on the COMSOL simulation dataset and observe consistent improvements over all baselines across amplitude, phase, spectral, and energy-based metrics, demonstrating that our frequency-domain modeling generalizes well to fully synthetic volumetric acoustic fields.

| Method | Amp | Ang | Spec | STFT | Ene. | Env. | T60 | EDT |
|---|---|---|---|---|---|---|---|---|
| INRAS | 1.28 | 1.60 | **2.18** | 4.11 | 2.95 | 12.88 | 14.6 | 51.58 |
| NAF | 1.53 | 1.61 | 2.76 | 3.53 | 5.40 | 19.41 | 29.09 | 102.88 |
| AVR | 0.81 | 1.60 | 2.07 | 5.01 | 3.12 | 17.08 | 26.36 | 35.14 |
| INFER | **0.78** | **1.60** | 2.42 | **3.10** | **2.75** | **7.2** | **12.9** | **29.4** |

Table 7: Evaluation on the COMSOL dataset.

- Per-Frequency Evaluation Across Third-Octave Bands: We further report per-frequency performance across third-octave bands. INFER achieves the lowest magnitude and phase error in nearly every band, including both low-frequency (100–400 Hz) and high-frequency ( greater than 2 kHz) regions. This demonstrates that the proposed frequency-domain formulation accurately models both global low-frequency structure and fine high-frequency phase behavior. Competing methods show pronounced degradation at higher frequencies, whereas INFER maintains stable performance across the full spectrum.

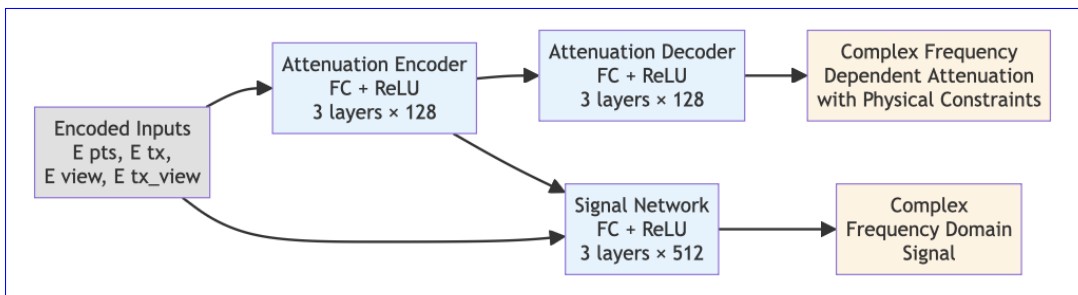

Figure 5: A visualization of our network architecture.

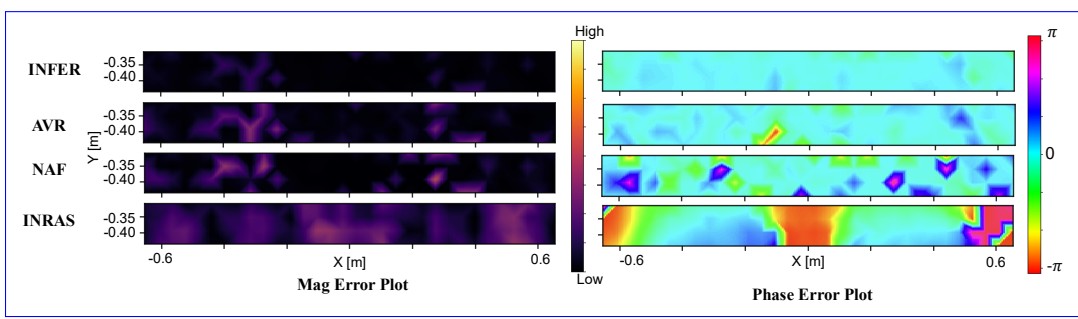

Figure 6: Spatial distribution of magnitude and phase reconstruction errors across the measurement plane for all methods. INFER achieves consistently low and spatially uniform errors, whereas baseline methods exhibit localized error concentrations and phase discontinuities.

### A.4 RAY-MARCHING

This section provides additional details on the ray-marching procedure used in INFER, addressing the reviewer's request for clarification. Our approach follows a deterministic, forward-integration scheme that accumulates complex transmittance and directional transfer responses along receiver-centered rays.

**Ray Initialization and Sampling Pattern:** Rays are cast *from the receiver position* $\mathbf{p}_r$ over a fixed azimuth–elevation grid. We use a uniformly sampled spherical grid (typically $N_\theta \times N_\phi$ directions), which defines a set of ray directions

$$\mathcal{R} = \{\, \mathbf{d}_i \mid i = 1, \ldots, N_{\text{rays}} \,\}.$$

Importantly, rays are not required to converge to any specific scene point $\mathbf{p}$; instead, each ray explores one possible propagation path toward the receiver.

Along each ray direction, we perform fixed-step marching:

$$\mathbf{x}_{k+1} = \mathbf{x}_k + \Delta u \, \mathbf{d}_i, \qquad k = 0, 1, \ldots$$

where $\Delta u$ is the spatial step size. At each step $\mathbf{x}_k$ we query the neural field to obtain the frequency-dependent attenuation and directional response needed for the recursive transmittance update.

**No Secondary Rays or Path Branching:** INFER does *not* spawn new rays at intermediate points. Although many acoustic simulators rely on explicit secondary rays to model reflections, INFER learns frequency-dependent attenuation and re-transmission fields that implicitly encode multi-bounce behavior. This greatly simplifies the ray geometry while still capturing rich propagation effects through the learned neural field.

**Ray Culling and Termination:** To improve efficiency, rays are terminated early when the accumulated amplitude becomes negligible. Specifically, if

$$\big\| T_k(f) \big\| < \epsilon,$$

Table 8: Per-frequency evaluation across third-octave bands. Lower is better for both metrics.

| | Magnitude Error | | | | Phase Error | | | |
|---|---|---|---|---|---|---|---|---|
| Freq (Hz) | INFER (Ours) | AVR | INRAS | NAF | INFER (Ours) | AVR | INRAS | NAF |
| 106 | **0.050** | 0.109 | 0.209 | 0.211 | **0.125** | 0.152 | 0.199 | 0.319 |
| 129 | **0.041** | 0.068 | 0.197 | 0.215 | **0.089** | 0.138 | 0.380 | 0.435 |
| 199 | **0.129** | 0.392 | 1.015 | 0.207 | **0.044** | 0.127 | 0.084 | 0.122 |
| 316 | **0.139** | 0.302 | 0.656 | 0.259 | **0.103** | 0.223 | 1.413 | 0.379 |
| 398 | **0.199** | 0.500 | 0.980 | 0.340 | **0.081** | 0.197 | 0.695 | 0.238 |
| 504 | **0.108** | 0.198 | 0.472 | 0.239 | **0.143** | 0.234 | 0.850 | 0.431 |
| 633 | **0.086** | 0.238 | 0.686 | 0.206 | **0.054** | 0.122 | 0.797 | 0.350 |
| 797 | **0.136** | 0.249 | 0.380 | 0.189 | **0.106** | 0.167 | 0.935 | 0.317 |
| 996 | **0.118** | 0.204 | 0.429 | 0.186 | **0.175** | 0.258 | 1.223 | 0.456 |
| 1,254 | **0.116** | 0.214 | 0.310 | 0.148 | **0.174** | 0.344 | 1.318 | 0.593 |
| 1,606 | **0.103** | 0.155 | 0.240 | 0.144 | **0.262** | 0.341 | 1.244 | 0.382 |
| 2,004 | **0.089** | 0.133 | 0.333 | 0.144 | **0.281** | 0.402 | 1.488 | 0.426 |
| 2,496 | **0.064** | 0.079 | 0.097 | 0.135 | **0.373** | 0.414 | 1.547 | 0.381 |
| 3,152 | **0.112** | 0.233 | 0.249 | 0.124 | 0.414 | 0.672 | 1.335 | **0.358** |

for all considered frequencies (we use $\epsilon = 10^{-4}$ by default), the ray is culled. We also terminate rays after a maximum traversal distance equal to the bounding volume of the scene.

In summary:

- Rays originate from the receiver and fan out over a spherical grid.
- No rays are required to converge to particular points; instead, each ray samples and integrates complex contributions along its path.
- No secondary rays are spawned; multi-bounce effects are captured implicitly through the learned attenuation and retransmission fields.
- Rays are culled when transmittance falls below a threshold.

### A.5 Loss Functions and Weights

The total training objective is composed of multiple terms designed to supervise the model's output across spectral amplitude, phase, energy structure, and physical consistency. The overall loss is expressed as:

$$L_{\text{total}} = \lambda_{\text{spec}} L_{\text{spec}} + \lambda_{\text{mag}} L_{\text{mag}} + \lambda_{\text{phase}} L_{\text{phase}} + \lambda_{\text{env}} L_{\text{env}} + \lambda_{\text{energy}} L_{\text{energy}} + \lambda_{\text{KK}} L_{\text{KK}} + \lambda_{\text{STFT}} L_{\text{MR-STFT}}. \tag{13}$$

In our experiments, we set $\lambda_{\text{spec}} = 16$, $\lambda_{\text{mag}} = 4$, $\lambda_{\text{phase}} = 1$, $\lambda_{\text{env}} = 0.25$, $\lambda_{\text{energy}} = 2$, $\lambda_{\text{KK}} = 0.25$, and $\lambda_{\text{STFT}} = 0.25$. Beyond these global weights, we apply frequency-dependent weighting: for $L_{\text{phase}}$ we emphasize low and mid frequencies by setting $w(f) = 1.2$ up to 1.5 kHz (125 bins), $w(f) = 1$ until 5 kHz (425 bins), and then smoothly tapering to 0.8 across the log-frequency axis; for $L_{\text{mag}}$, weights are 1 up to 1.5 kHz, 1.25 until 5 kHz, and then tapered to 0.8; and for $L_{\text{spec}}$, we assign 1.25 up to 5 kHz before tapering. These schedules follow psychoacoustics informed weighting strategies, prioritizing perceptually critical bands while de-emphasizing unreliable high-frequency bins.

### A.6 Training Pipeline

We use the script `avr_runner_complex_FD_kk.py` for training. Each step involves:

1. Ray-based spherical integration using normalized receiver and transmitter coordinates.
2. Prediction of complex signals and attenuation fields.
3. Loss computation including frequency-weighted spectrum and KK regularization.

4. Gradient backpropagation with NaN filtering and norm clipping.

5. Mixed precision training and GPU memory optimization.

**Special Considerations**:

- Gradient clipping to max norm 1.
- Automatic mixed precision (AMP) to save memory.
- Complex loss handling via separate $\Re[H(f)]$ and $\Im[H(f)]$ paths.
- KK regularizer computed using discrete Hilbert transform with frequency masking and tapering.

## A.7 REPRODUCIBILITY CHECKLIST

**Software Environment**:

- `Python 3.8`, `PyTorch 1.12.0`, `CUDA 11.6`
- `tinycudann 1.6`, `auraloss 0.4.0`, `tensorboard 2.8.0`

**Training Script (Single GPU)**:

```
python avr_runner_complex_FD_kk.py \
  --config config_files/avr_buck_complex_dir_FD.yml \
  --model_type AVRModel_complex_FD_FreqDep_PhaseCorrection \
  --renderer_type AVRRenderFD_FreqDep_PhaseCorrection_KK \
  --batchsize 1 \
  --dataset_dir /path/to/dataset
```

## A.8 HARDWARE REQUIREMENTS

**Minimum**:

- GPU: NVIDIA RTXA6000 (10GB+ VRAM)

**Recommended**:

- GPU: L40S

Training time is approximately 24 hours on a single L40S.

## A.9 EVALUATION METRICS

All metrics reported in Tables 1, 3, and 4 represent **Mean Absolute Error (MAE)** unless otherwise specified. Lower values indicate better performance across all metrics.

### FREQUENCY-DOMAIN METRICS

**Envelope Error.** Given the time domain ground truth impulse response $h^*[n]$ and our prediction $h[n]$, we compute the envelope error by first obtaining the envelope using the Hilbert transform to get the analytic signal and then applying the absolute value, as follows:

$$\text{Env}^* = |\text{Hilbert}(h^*)| \tag{14}$$

The normalized *envelope error* is defined as follows (we multiply by 100 to report as percentage):

$$\text{Envelope error} = 100 \times \text{Mean}\left(\frac{|\text{Env}^* - \text{Env}|}{\max(\text{Env}^*)}\right) \tag{15}$$

**Phase and Amplitude Error.** Given the frequency domain ground truth impulse response $H^*[f]$ and our prediction $H[f]$, we use cosine and sine functions to quantify the *phase error*:

$$\text{Phase error} = \text{Mean}(|\cos(\angle H^*) - \cos(\angle H)| + |\sin(\angle H^*) - \sin(\angle H)|) \tag{16}$$

The *amplitude error* is defined as:

$$\text{Amplitude error} = \text{Mean}\left(\frac{|abs(H^*) - abs(H)|}{abs(H^*)}\right) \tag{17}$$

**Spectral Error (Spec).** Mean absolute difference between real and imaginary components:

$$\text{Spec} = \text{Mean}(|\Re[H^*] - \Re[H]| + |\Im[H^*] - \Im[H]|) \tag{18}$$

TIME-DOMAIN METRICS

**Multi-Resolution STFT Loss (STFT).** We compute the multi-resolution spectral distance using multiple STFT window sizes following Yamamoto et al. (2020):

$$L_{\text{MR-STFT}} = \frac{1}{M} \sum_{m=1}^{M} \left(L_{\text{sc}}^{(m)} + L_{\text{mag}}^{(m)}\right) \tag{19}$$

where $L_{\text{sc}}$ is the spectral convergence loss and $L_{\text{mag}}$ is the log-magnitude loss, computed over $M$ different FFT sizes.

**Energy Error (Ene.).** Cumulative energy deviation in the frequency domain:

$$\text{Energy error} = \text{Mean} \left| \sum_f |H^*[f]|^2 - \sum_f |H[f]|^2 \right| \tag{20}$$

PERCEPTUAL ACOUSTIC METRICS

**T60 Reverberation Time.** The T60 metric measures the time required for sound pressure level to decay by 60 dB after the source stops. We compute the MAE between predicted and ground truth T60 values in milliseconds. Lower error indicates better preservation of room decay characteristics.

**Early Decay Time (EDT).** EDT measures the initial decay rate (first 10 dB) and is particularly important for perceived spaciousness. We report MAE in milliseconds.

**Clarity C50 (dB).** The C50 metric quantifies speech intelligibility as the ratio of early (0-50ms) to late energy. We report MAE in decibels.

PER-FREQUENCY ANALYSIS

For Table 4, we report frequency-band-specific MAE computed at center frequencies of third-octave bands (180 Hz, 360 Hz, 720 Hz, 1440 Hz, 2880 Hz) for both magnitude and phase errors. This allows fine-grained analysis of model performance across the audible spectrum.

A.10 AUDIO HARDWARE SPECIFICATION

We detail here the acoustic transducer setup used for our data collection in the *BUCK* testbed and the production *Tesla Model X* vehicle. Both systems were equipped with a rich spatial arrangement of loudspeakers and a high-fidelity microphone array to facilitate spatial audio capture and reconstruction.

**Speaker Configuration.** While the Tesla Model X uses the default speakers, In BUCK, the active speakers used for sound excitation include:

- **Center Dash Speaker:** A 3.5-inch wideband driver, such as the SLA Ram3 or Dayton Audio DMA90-4, capable of full-range output from $85.00\,\text{Hz}$ to $20.00\,\text{kHz}$. In typical configurations, these are high-passed at approximately $100.00\,\text{Hz}$ to avoid low-frequency distortion.

- **Rear Door Speakers:** Morel Tempo Ultra Integra 402 or 602 coaxial hybrids with wideband support ($55.00\,\text{Hz}$ to $22.00\,\text{kHz}$), high sensitivity, and power handling up to $120.00\,\text{WRMS}$. These speakers internally crossover between woofer and tweeter around $2.50\,\text{kHz}$–$3.00\,\text{kHz}$.

- **Rear Height Speakers:** Tang Band T2-2136SF full-range modules and Morel CCWR254 midrange drivers, mounted in ceiling/rear hatch positions to introduce vertical spatial content, spanning $80.00\,\text{Hz}$ to $20.00\,\text{kHz}$. Crossover filters are typically applied around $800.00\,\text{Hz}$–$1.00\,\text{kHz}$ depending on system design and companion driver.

This layout approximates a 7.1.4 immersive audio setup and enables extensive sampling of reverberant and directional field responses across both testbeds.

**Microphone Array.** We use the commercially available **MiniDSP UMA-16** USB microphone array, which offers 16 omnidirectional MEMS microphones in a linear array form factor. This array supports high-resolution spatial sampling across the cabin, enabling dense reconstruction of directional impulse responses.

### A.11 BASELINE METHODS

To rigorously evaluate the effectiveness of our proposed system INFER, we compare against three representative baselines, each reflecting a different class of acoustic modeling approach:

- **AVR**: A hybrid time–frequency domain neural field that learns time-domain impulse responses via a differentiable renderer. While AVR applies frequency-domain path delays in its rendering, the model is supervised in the time domain and does not explicitly learn frequency-dependent attenuation or dispersion.

- **NAF (Neural Acoustic Field)**: A neural field trained directly in the time domain using MSE and time-domain perceptual losses. NAF ignores frequency-domain supervision and is evaluated primarily on time-domain waveform fidelity.

- **INRAS (Impulse Response as Signal)**: A signal regression approach where the model directly regresses to the complex impulse response waveform as a 1D signal. INRAS uses STFT-based perceptual loss, but it does not exploit any spatial priors or directional conditioning.

Each baseline is re-implemented in our codebase with their respective loss functions and evaluation metrics faithfully reproduced, using the same training datasets, preprocessing pipelines, and neural architecture backbones where applicable.

### A.12 TRAINING CONFIGURATION

All baselines are trained with identical configurations to ensure fair comparisons:

- **Dataset:** We use the same training/validation/test splits from our real (BUCK) and synthetic (Tesla) datasets for all methods.

- **Resolution:** The frequency bins, spatial sampling resolution, and directional integration are matched across all methods.

- **Training Epochs:** All models are trained for 500 epochs with early stopping based on validation loss.

- **Batch Size:** Batch size of 1 is used due to GPU memory constraints, consistent with prior volumetric rendering works.

- **Evaluation:** All comparisons are evaluated on both magnitude and phase accuracy across the frequency range of interest, in addition to perceptual STFT loss and energy-based metrics.

### A.13 Loss Function Implementation

**AVR.** We follow the original AVR formulation and use the same set of losses described in the paper.

**NAF.** The NAF baseline is trained using the standard losses introduced in its original work, without any additional frequency-domain regularization.

**INRAS.** For INRAS, we adopt the exact losses specified in the original paper, without modification.

### A.14 Architectural Modifications

To isolate the effects of spectral supervision and renderer formulation, all baseline models are built upon the same backbone MLP architecture as our method:

- 8-layer fully connected network with sinusoidal positional encoding.
- Input: $(\mathbf{p}_{\text{tx}}, \hat{\mathbf{n}}_{\text{tx}}, \mathbf{x}, \hat{\mathbf{n}})$ with appropriate frequency and spatial encodings.
- Output: Real-valued waveform or complex spectrum depending on method.

### A.15 Notes on Fairness and Robustness

To ensure fairness in evaluation:

- All models are trained with the same GPU hardware, random seed initialization, and PyTorch version.
- We use the same optimizer (Adam) and learning rate schedule across all models unless otherwise noted.
- All baselines are evaluated using our standardized renderer and metric pipeline to eliminate post-processing inconsistencies.
- We tune loss weights and learning rates for each baseline to ensure their best performance under our training conditions.

Overall, our comparison demonstrates that INFER substantially outperforms these baselines across spectral and perceptual metrics due to its physics-informed supervision and frequency-aware modeling.

