# OpenReview forum: "INFER : Learning Implicit Neural Frequency Response Fields for Confined Car Cabin"
_ICLR.cc/2026/Conference — Submitted to ICLR 2026_

### Official Review · Reviewer_qS5V · 2025-10-31

**Soundness:** 3
**Presentation:** 3
**Contribution:** 2
**Rating:** 6
**Confidence:** 2

**Summary:**

Authors introduced INFER, a novel spectral-domain framework that models acoustic propagation in
confined environments using implicit neural representations. By operating directly in the frequency
domain, their method enables perceptually grounded supervision, hardware-aware weighting, and
physically consistent regularization through the Kramers–Kronig constraint. They did extensive evaluations on real and simulated car cabin datasets demonstrate that INFER substantially outperforms prior time-domain and
hybrid approaches, achieving over 50% improvement in phase accuracy and 39% in magnitude fidelity relative to the best baseline.

**Strengths:**

INFER substantially outperforms prior time-domain and hybrid approaches, achieving over 50% improvement in phase accuracy and 39% in magnitude fidelity relative to the best baseline.

They  first propose to encode KK-consistent complex attenuation in a neural acoustic renderer, preventing non-physical phase behavior and improving both interpretability and generalization.

Their proposed total loss is novel and it is a key for achieving the best accuracy.

**Weaknesses:**

Authors are solving a specific problem of neural modeling for frequency response field for confined car cabins.
Can this approach be scaled on other applications?

**Questions:**

It is surprising that this approach, based on vanilla sequence of fully connected layers is outperforming other methods.
Please explain why?
E.g. that means that the baseline is weak, or method in this paper uses additional data which are not used by the baseline?

---

> ### Author Response · Authors · 2025-11-20
> **Additional Room Scale Experiments and Ablation Studies**
>
> We sincerely thank Reviewer qS5V for aptly identifying  and recognition of INFER's contributions. We appreciate the acknowledgment of our performance improvements, novel KK-consistent modeling, and loss formulation. We address the scalability question and architectural design below.
>
> ### **W1: Scalability Beyond Car Cabins**
>
> **We completely agree with this concern and have now demonstrated INFER's broader applicability.** To address the concern about scope, we have added experiments on three publicly available **room-scale datasets**: MeshRIR , RAF-Furnished, and RAF-Empty (measured rooms).
>
> # Table 2: Evaluation on room-scale environments.
>
> | Method | MeshRIR |  |  |  |  |  | RAF-Furnished |  |  |  |  |  | RAF-Empty |  |  |  |  |  |
> |--------|---------|------|------|------|------|------|---------|------|------|------|------|------|---------|------|------|------|------|------|
> |  | Phase | Amp. | Env. | T60 | C50 | EDT | Phase | Amp. | Env. | T60 | C50 | EDT | Phase | Amp. | Env. | T60 | C50 | EDT |
> | AAC-nearest | 1.47 | 0.91 | 1.40 | 8.6 | 2.20 | 58.8 | 1.60 | 1.09 | 4.83 | 13.0 | 3.41 | 73.5 | 1.60 | 1.09 | 4.83 | 13.0 | 3.41 | 73.3 |
> | AAC-linear | 1.44 | 0.89 | 1.42 | 8.2 | 2.29 | 58.9 | 1.60 | 0.99 | 3.81 | 12.4 | 3.65 | 90.2 | 1.59 | 1.10 | 5.22 | 13.1 | 3.25 | 71.5 |
> | Opus-nearest | 1.45 | 0.72 | 1.37 | 5.2 | 1.26 | 35.7 | 1.60 | 1.19 | 5.35 | 14.4 | 3.78 | 80.3 | 1.59 | 1.16 | 4.58 | 13.3 | 4.25 | 100.6 |
> | Opus-linear | 1.43 | 0.69 | 1.37 | 6.9 | 1.83 | 49.3 | 1.60 | 1.47 | 5.74 | 13.1 | 3.55 | 77.8 | 1.59 | 0.95 | 4.26 | 12.7 | 3.94 | 95.5 |
> | NAF | 1.61 | 0.64 | 1.59 | 4.2 | 1.25 | 39.0 | 1.62 | 0.93 | 5.34 | 7.1 | **0.98** | **20.6** | 1.62 | 0.85 | 4.67 | 8.0 | 1.22 | 26.3 |
> | INRAS | 1.61 | 0.77 | 1.85 | 3.4 | 1.47 | 40.7 | 1.62 | 0.96 | 6.43 | 6.9 | 1.08 | 21.4 | 1.62 | 0.88 | 4.72 | 7.6 | 1.21 | 25.8 |
> | AVR | 1.48 | 0.54 | **1.15** | 3.9 | 0.92 | 35.1 | **1.58** | 0.28 | 5.79 | 6.6 | 1.12 | 22.88 | **1.58** | 0.29 | 5.16 | 6.3 | 1.18 | 24.3 |
> | INFER (without KK) | 1.224 | 0.26 | 7.40 | 2.8 | 0.57 | 127.1 | **1.58** | **0.2337** | 5.33 | 6.35 | 1.15 | 22.56 | **1.58** | **0.23** | 4.73 | 6.0 | 1.07 | 23.1 |
> | INFER | **1.194** | **0.24** | 7.34 | **3.14** | **0.50** | **12.45** | **1.58** | **0.2197** | **5.40** | **6.34** | **1.08** | **22.17** | **1.58** | **0.23** | **4.76** | **6.3** | **1.11** | **23.5** |
>
>
> On all three, INFER achieves the best or tied-best  across methods. We also show that INFER remains superior to INFER without KK consistency regularizer variant model to prove the effectiveness of the regularization. The table has also been added to sec 5.3 Table 2.
>
> ### Q1. “It is surprising that this approach, based on vanilla sequence of fully connected layers is outperforming other methods. Please explain why? E.g. that means that the baseline is weak, or method in this paper uses additional data which are not used by the baseline?”
>
> We have carefully designed our setup so that all methods use exactly the same data. Every baseline (AAC/Opus interpolation, NAF, INRAS, AVR) and INFER is trained and evaluated on the *same* set of measured acoustic responses for each dataset. Train/validation/test splits are shared across methods and are spatially disjoint at the receiver level (test receivers are never seen in training).
>
> **Strength of baselines.**
>
> In our experiments, the strongest prior neural field baselines already improves substantially over interpolation and codec baselines, especially in magnitude metrics. INFER’s improvements are on top of these strong baseline. The superiority of INFER comes from an end-to-end frequency  domain differentiable forward modeling, and new supervision terms introduced.
>
> We hope we addressed all the concerns raised by the reviewer and hope the reviewer will consider raising their score given additional evaluation on real world datasets that strengthen INFER’s generalizability claim.

---

> > ### Author Response · Authors · 2025-11-25
> > **Follow-up on author response**
> >
> > Dear Reviewer qS5V,
> > Thanks again for your feedback regarding our paper. We have responded to your initial comments/questions and have incorporated them accordingly into our revised manuscript.
> >
> > We would greatly appreciate you taking a moment to review our response, reassess our revised manuscript from all aspects, share your thoughts, and update your score accordingly. We sincerely believe that our paper has been strengthened a lot thanks to the feedback. We look forward to hearing from you.

---

### Official Review · Reviewer_dpmJ · 2025-10-31

**Soundness:** 3
**Presentation:** 3
**Contribution:** 2
**Rating:** 2
**Confidence:** 4

**Summary:**

This paper introduces Implicit Neural Frequency Response fields (INFER), a neural acoustic field learning framework for confined car cabin.

Key contributions are:
- First to apply the end-to-end frequency domain modeling for car cabins.
- Proposed Kramers–Kronig physical consistency regularization to enforce spectral attenuation and phase delay across frequencies.
- Evaluation on both COMSOL simulated data and real car cabins show that INFER outperforms the other baselines (NAF, INRAS, and AVR).

**Strengths:**

- **Originality:**

    INFER is the first to offer fully frequency-domain neural acoustic field modeling for car cabins.
    Integrating Kramers–Kronig grounded physics in a regularization is novel and well-motivated.

- **Quality:**

    The technical presentation is thorough, underlying principles are detailed, and the method is carefully justified physically and psychoacoustically.
    The evaluations on both synthetic and real environments shows fair benchmark comparisons. Multiple metrics are reported and samples are visualized to support claims.

- **Clarity:**

    The paper generally presents its methodology clearly. While there are some omissions, it provides relatively detailed figures, including hyperparameters.

- **Significance:**

    *If the author releases the code and data* as promised, this study will be significant as a benchmark. Yet, this reviewer cannot find neither code nor data from the supplementary material at this moment.

**Weaknesses:**

1. **Novelty:**

    The method builds on existing ideas in neural implicit fields for acoustics (NAF, INRAS, AVR).
    While the application of frequency-domain learning and KK regularization is novel *for car acoustics*, such novelty, as suggested by its distinctions from prior research, is somewhat limited.
    The frequency-domain modeling is not actually a particularly unique method and has already been applied in many other spatial audio studies [1,2]. As the author also explained, the fact that the frequency-time hybrid approach AVR already exists is in a similar vein. Reports of AVR's inferior performance compared to NAF make one wonder again whether the advantages of frequency-domain modeling truly is significant in this context. For this reason, this reviewer cannot agree with the authors' claim that the 'end-to-end frequency-domain forward model' is novel, as insisted in the key contributions.

2. **Ablation study:**

    The paper only reports on the proposed approach for the specific case, which is of a car cabin. This alone does not undermine the author's claim, but it does imply that the audience of interested readers may be somewhat limited. Without ablation studies, it is hard to provide meaningful insights unless one is specifically interested in car cabins. For instance, it remains unclear whether frequency modeling or the KK regularization trick also benefits learning indoor acoustic fields (where NAF, INRAS, and AVR were tested on). Therefore, the author's claim is valid only for ‘car cabins’.

3. **Generalization:**

    The datasets (COMSOL, BUCK, Tesla Model X) are well chosen, and providing details about the measurement setup is much appreciated.
    However, it is extremely difficult to determine how many source-receiver pairs were sampled per scene and where they were sampled.
    To my understanding, like many other neural acoustic field learning works, this paper also tackles the problem of generalization within a scene (not across scenes). In this case, it is critical to see how the approach generalizes well for the training data's sparsity and its sampling distribution, as the key issue is "how sparse the source-receiver pairs used for training can be".
    Yet, there are no experiments addressing this at all.

4. **Interpretability and Physical Plausibility:**

    The paper enforces KK relationships in loss, but the qualitative impact on physical interpretability is mostly assumed.
    (For example, do reconstructions violate causality if KK is removed? Do phase/magnitude predictions by INFER become less plausible without such regularization?)
    More ablation studies on the necessity of KK, and analysis of physical interpretability (not just metric fidelity), would strengthen the claims.

5. **Baseline Fairness:**

    The authors explain that all baselines are re-implemented in their codebase. The effort is to be commended, but there could be unintended pitfalls.
    For example, the baseline was not properly verified, and it is questionable that NAF—which appeared first among NAF, INRAS, and AVR, and has generally been reported to perform worse than INRAS or AVR—shows the second-best performance after INFER.
    It is unclear whether the automotive cabin environment is exceptionally unique, as there is no way to know (since no comparative experiments were conducted).

[1] Lee, J. W., & Lee, K. (2023). Neural fourier shift for binaural speech rendering. In ICASSP 2023-2023 IEEE International Conference on Acoustics, Speech and Signal Processing (ICASSP). IEEE.

[2] Di Carlo, D., Nugraha, A. A., Fontaine, M., Bando, Y., & Yoshii, K. (2024). Neural Steerer: Novel steering vector synthesis with a causal neural field over frequency and direction. In 2024 IEEE International Conference on Acoustics, Speech, and Signal Processing Workshops (ICASSPW) (pp. 740-744). IEEE.

[3] Wang, M. L., Sawata, R., Clarke, S., Gao, R., Wu, S., & Wu, J. (2024). Hearing anything anywhere. In Proceedings of the IEEE/CVF Conference on Computer Vision and Pattern Recognition (pp. 11790-11799)

**Questions:**

1. Regarding Appendix A7 (Evaluation Metrics), it seems like KK Violation Metric was used, but this metric does not appear anywhere in the paper. Please report this metric?

2. The paper states that “hardware-aware weighting” is a key contribution, but nowhere does it explain *where it originated from* or *how the values were determined*. From reading section 4.3, it seems like $w(f)$ is what the authors mean by "hardware-aware weighting," which turns out to be a heuristic "frequency-dependent weighting" in Appendix A3. If this actually improves performance, an ablation study should be included. For example, it should report how much performance drops when this trick is omitted from the proposed methodology, or whether applying this trick to the baseline architecture yields the same performance gains. (The same applies to the KK loss function.)

3. Although the content is repeatedly mentioned under ‘Weakness,’ is the proposal in this paper only applicable to car cabins? Are there no experimental results conducted in a room? Which part of the proposal specifically functions for the car cabins?

4. When will the code and dataset be made available as stated?

---

> ### Author Response · Authors · 2025-11-20
> **Clarifications on Novelty, Additional Room Scale Experiments, and Ablation Studies**
>
> We thank Reviewer dpmJ for the recognition of our technical presentation, physical grounding, and comprehensive evaluation. We address each concern below and highlight new experimental results that significantly strengthen our claims.
>
> ### **W1: Novelty of frequency-domain modeling**
>
> We respectfully disagree with the assessment that frequency-domain modeling is “not particularly unique” in the context of *frequency response field* reconstruction. Our contribution is not simply to apply spectral processing, but to introduce a neural *frequency response field* with three key differences from prior work:
>
> 1. **Frequency response field vs. spectral processing of IRs.**
>
>     The cited works [1,2] and prior neural acoustic fields (NAF, INRAS, AVR) ultimately predict **time-domain impulse responses** and only use spectral quantities for losses or feature extraction. In contrast, INFER directly parameterizes and predicts a **continuous complex frequency response field** (H(f,\mathbf{x})) as the primary representation, conditioned on source–receiver pose. This allows us to interpolate and manipulate frequency responses at arbitrary spatial locations without ever reverting to a time-domain IR. We will clarify in the paper that our novelty claim is specifically about modeling *frequency response fields* with coordinate-based networks, not about using the Fourier domain per se.
>
> 2. **End-to-end learning of frequency-specific attenuation and retransmission.**
>
>     AVR uses a hybrid time/frequency forward model but still assumes frequency-independent attenuation and produces time-domain IRs. INFER instead learns **frequency-dependent complex attenuation (\delta(f,\mathbf{x}))** and **directional retransmission (S(f,\mathbf{x},\hat{\mathbf{n}}))** directly in the neural field, with KK-consistent coupling between amplitude and phase. Each frequency bin is modeled independently, enabling per-frequency weighting (down-weighting unstable crossover bands), perceptual, frequency-selective phase supervision at low frequencies, and direct interpolation of frequency responses in space for arbitrary queries.
>
> 3. **Relevance of frequency response fields for industrial audio.**
>
>     In practical automotive pipelines (EQ, beamforming, ANC, spatial renderers), engineers work directly with *frequency responses* rather than raw IRs. Our representation therefore mirrors the native objects used for loudspeaker tuning and beamformer desig. Even a bandwidth-limited frequency response field is immediately useful for tasks such as car-cabin equalization and robust voice pickup.
>
>
> Regarding the comparison to AVR, our new experiments on *room-scale* datasets (MeshRIR, RAF-Furnished, RAF-Empty) show that AVR is often stronger than NAF in these larger environments, while INFER still matches or outperforms both across metrics. In confined, highly resonant car cabins, however, AVR degrades sharply whereas INFER remains stable. This suggests that the limitation is not frequency-domain modeling itself, but the particular hybrid design and frequency-independent attenuation assumptions of AVR, and it further supports our claim that explicitly learning frequency response fields with frequency-specific attenuation is especially beneficial in confined spaces.
>
> We have updated the manuscript to (i) explicitly acknowledge prior uses of frequency-domain processing [1,2], (ii) state our novelty  as “a spectral-domain neural *frequency response field* with KK-consistent complex attenuation and perceptual/hardware-aware spectral modeling,” and (iii) tone down wording around “first” while retaining a clear, well-differentiated contribution statement.

---

> > ### Author Response · Authors · 2025-11-20
> >
> > ### **W2: Ablation Studies and Generalization Beyond Car Cabins**
> >
> > We now conduct experiments on standard room-scale environments (MeshRIR, RAF-Furnished, RAF-Empty) to demonstrate that INFER generalizes beyond car cabins; results appear in Table 2.
> >
> > **Key Findings:**
> >
> > 1. INFER generalizes well to room environments, achieving best or competitive performance across metrics.
> > 2. The frequency-domain approach provides benefits even in less confined spaces.
> > 3. KK regularization is most critical in highly resonant cabins and less influential in room-scale scenes.
> >
> > # Table 2: Evaluation on room-scale environments.
> >
> > | Method | MeshRIR |  |  |  |  |  | RAF-Furnished |  |  |  |  |  | RAF-Empty |  |  |  |  |  |
> > |--------|---------|------|------|------|------|------|---------|------|------|------|------|------|---------|------|------|------|------|------|
> > |  | Phase | Amp. | Env. | T60 | C50 | EDT | Phase | Amp. | Env. | T60 | C50 | EDT | Phase | Amp. | Env. | T60 | C50 | EDT |
> > | AAC-nearest | 1.47 | 0.91 | 1.40 | 8.6 | 2.20 | 58.8 | 1.60 | 1.09 | 4.83 | 13.0 | 3.41 | 73.5 | 1.60 | 1.09 | 4.83 | 13.0 | 3.41 | 73.3 |
> > | AAC-linear | 1.44 | 0.89 | 1.42 | 8.2 | 2.29 | 58.9 | 1.60 | 0.99 | 3.81 | 12.4 | 3.65 | 90.2 | 1.59 | 1.10 | 5.22 | 13.1 | 3.25 | 71.5 |
> > | Opus-nearest | 1.45 | 0.72 | 1.37 | 5.2 | 1.26 | 35.7 | 1.60 | 1.19 | 5.35 | 14.4 | 3.78 | 80.3 | 1.59 | 1.16 | 4.58 | 13.3 | 4.25 | 100.6 |
> > | Opus-linear | 1.43 | 0.69 | 1.37 | 6.9 | 1.83 | 49.3 | 1.60 | 1.47 | 5.74 | 13.1 | 3.55 | 77.8 | 1.59 | 0.95 | 4.26 | 12.7 | 3.94 | 95.5 |
> > | NAF | 1.61 | 0.64 | 1.59 | 4.2 | 1.25 | 39.0 | 1.62 | 0.93 | 5.34 | 7.1 | **0.98** | **20.6** | 1.62 | 0.85 | 4.67 | 8.0 | 1.22 | 26.3 |
> > | INRAS | 1.61 | 0.77 | 1.85 | 3.4 | 1.47 | 40.7 | 1.62 | 0.96 | 6.43 | 6.9 | 1.08 | 21.4 | 1.62 | 0.88 | 4.72 | 7.6 | 1.21 | 25.8 |
> > | AVR | 1.48 | 0.54 | **1.15** | 3.9 | 0.92 | 35.1 | **1.58** | 0.28 | 5.79 | 6.6 | 1.12 | 22.88 | **1.58** | 0.29 | 5.16 | 6.3 | 1.18 | 24.3 |
> > | INFER (without KK) | 1.224 | 0.26 | 7.40 | 2.8 | 0.57 | 127.1 | **1.58** | **0.2337** | 5.33 | 6.35 | 1.15 | 22.56 | **1.58** | **0.23** | 4.73 | 6.0 | 1.07 | 23.1 |
> > | INFER | **1.194** | **0.24** | 7.34 | **3.14** | **0.50** | **12.45** | **1.58** | **0.2197** | **5.40** | **6.34** | **1.08** | **22.17** | **1.58** | **0.23** | **4.76** | **6.3** | **1.11** | **23.5** |
> >
> > Table 3 reports ablations over loss terms, sampling strategy, and training sparsity, showing each component’s contribution and robustness to reduced data.
> >
> > # Table 3: Model ablations. Performance for the model variants.
> >
> > | Study Objectives | Variation | Phase | Amp. | Env. | T60 | EDT |
> > |------------------|-----------|-------|------|------|-----|-----|
> > | Loss Component | w/o mag loss | 0.74 | 0.18 | 1.6 | 8.9 | 7.1 |
> > |  | w/o phase loss | 0.98 | 0.2 | 1.8 | 7.8 | 6.2 |
> > |  | w/o energy loss | 0.5 | **0.12** | 0.99 | 24.0 | 3.5 |
> > |  | w/o kk loss | 0.77 | 0.18 | 1.7 | 9.8 | 7.3 |
> > |  | w/o stft loss | 0.64 | 0.15 | 1.4 | 7.0 | 4.4 |
> > |  | w/o spec loss | 1.44 | 0.25 | 2.6 | **2.4** | 2.7 |
> > |  | w/o env loss | 0.57 | 0.13 | 1.2 | 5.7 | 3.9 |
> > |  | w/o frequency weighting | 0.55 | 0.13 | 1.2 | 2.8 | 3.4 |
> > |  | w/ all loss components | **0.48** | **0.12** | **0.95** | 9.8 | **2.6** |
> > | Training Data Reduction | 30% | 1.08 | 0.25 | 1.9 | 3.6 | 4.4 |
> > |  | 50% | 0.81 | 0.19 | 1.4 | **3.2** | 3.7 |
> > |  | 60% | 0.68 | 0.15 | 1.3 | 3.5 | 3.7 |
> > |  | 75% | **0.5** | **0.12** | **0.95** | 9.8 | **2.6** |
> > | Sampling Parameters | 32 × 16 rays, 64 points | 0.98 | 0.43 | 4.2 | 13.6 | 10.2 |
> > |  | 48 × 24 rays, 64 points | 0.91 | 0.24 | 1.9 | 10.06 | 6.1 |
> > |  | 64 × 32 rays, 64 points | **0.5** | **0.12** | **0.95** | 9.8 | **2.6** |
> > |  | 64 × 32 rays, 40 points | 1.13 | 0.32 | 2.3 | **7.0** | 6.9 |
> > |  | 64 × 32 rays, 70 points | **0.48** | **0.11** | **0.91** | 7.8 | 2.8 |
> >
> > ### **W3: Generalization on Sampling Positions and Sparsity**
> >
> > To address this concern, we added detailed dataset statistics in Appendix A.2 (Table 6). INFER is designed to generalize within a scene. Our sparsity ablation in Table 3 sub-samples training receivers at 30%, 50%, 60%, and 75% of the original positions; performance degrades smoothly as data become sparser, indicating robust generalization under varied sampling density.
> >
> > ### **W4: Interpretability and Physical Plausibility**
> >
> > In the revised version, we extend our analysis with a full ablation over all loss components, not only the KK regularizer (Table 3). Removing KK causes clear degradation across magnitude, phase, spectral, and energy metrics: phase error increases by ~60% and amplitude error by ~50%. Without KK, the network can fit data while violating the causal coupling between attenuation and dispersion, yielding less interpretable and physically inconsistent reconstructions. KK-regularized models preserve coherent magnitude–phase behavior and deliver consistent performance gains, also visible across datasets in Table 2.

---

> > > ### Author Response · Authors · 2025-11-20
> > >
> > > ### **W5: Baseline Fairness and NAF Performance**
> > >
> > > ---
> > >
> > > First, to validate our re-implementations, we evaluated NAF, INRAS, and AVR on **publicly available datasets**(MeshRIR, RAF-Empty, RAF-Furnished) and confirmed that they reproduce the qualitative trends and performance ranges reported in the original papers. This gives us confidence that our code paths for all baselines are correct.
> > >
> > > Second, in all experiments the baselines share the **same backbone MLP architecture** (depth, width, positional encodings) and training budget as INFER; the only differences are in the forward model and associated loss terms. For each method we **faithfully re-implemented its original losses and training protocol** and tuned them to their best validation performance under this common budget. We also use **identical train/val/test splits, preprocessing, and evaluation metrics** for all models.
> > >
> > > Regarding the observation that NAF appears as the second-best method in our results: this does not indicate an implementation bug but rather reflects our **different target regime**. Our task focuses on strongly confined car cabins (and small rooms), which differ substantially from the larger, less confined environments used in prior NAF/INRAS/AVR benchmarks. In this setting, we consistently observe that NAF’s simpler field parameterization is more robust than AVR/INRAS under a shared backbone, whereas INFER further improves both magnitude and phase metrics. We now state this explicitly in the main text and appendices to make clear that the change in ranking is a consequence of the **changed geometry and evaluation protocol**, not an unfair baseline implementation.
> > >
> > > ### **Q1: KK Violation Metric Reporting**
> > >
> > > KK Violation Metric can be calculated only for complex valued frequency domain attenuation modeling. As none of the past models output such attenuation models, it is infeasible to calculate the KK Violation Metric. We have removed KK violation metric from the list of metrics mentioned in A.7 section of the appendix. Instead we checked the effect of KK consistency regularizer
> > > ****
> > >
> > > ### **Q2: Hardware-Aware Weighting Details and Ablation**
> > >
> > > As noted in our response above, we now perform extensive ablations on both the KK loss and the frequency weighting w(f); the corresponding results are reported in Table 3. Removing either KK consistency regularizer or w(f) leads to a clear degradation across all metrics, indicating its benefit in INFER’s frequency-response–field formulation. Details on w(f) are available in Appendix Sec A.5.
> > >
> > > ### **Q3: Applicability Beyond Car Cabins**
> > >
> > > Addressed this comprehensively in W2 response above. **Summary:**
> > >
> > > **Yes, INFER works for standard rooms** (MeshRIR, RAF-Furnished, RAF-Empty results show best/competitive performance)
> > >
> > > **Specific advantages for car cabins:** For practical applications (automotive EQ, active noise cancellation, spatial audio rendering), practitioners work directly with frequency responses, not impulse responses. **Frequency response fields are the native representation for these tasks**
> > >
> > > **General advantages:** Perceptual supervision, frequency-specific modeling, physical consistency
> > >
> > > No architectural components are car-specific; the approach is universally applicable with configuration adjustments (e.g., reducing KK weight for rooms).
> > >
> > > ### **Q4: Code and Dataset Release**
> > >
> > > Code and a demo dataset has now been released  has been released at our project page: [https://anonymous1415510-spec.github.io](https://anonymous1415510-spec.github.io/)
> > >
> > > We hope we addressed all the concerns raised by the reviewer and hope the reviewer will consider raising their score given additional evaluation on real world datasets that strengthen INFER’s generalizability claim.

---

> > > > ### Author Response · Authors · 2025-11-25
> > > > **Follow-up on author response**
> > > >
> > > > Dear Reviewer dpmJ,
> > > > Thanks again for your feedback regarding our paper. We have responded to your initial comments/questions and have incorporated them accordingly into our revised manuscript.
> > > >
> > > > We would greatly appreciate you taking a moment to review our response, reassess our revised manuscript from all aspects, share your thoughts, and update your score accordingly. We sincerely believe that our paper has been strengthened a lot thanks to the feedback. We look forward to hearing from you.

---

### Official Review · Reviewer_fi6v · 2025-11-01

**Soundness:** 2
**Presentation:** 2
**Contribution:** 1
**Rating:** 2
**Confidence:** 3

**Summary:**

This paper presents INFER, a deep learning algorithm that predicts an acoustic transfer function at a 3D point. INFER is specially designed for environments with non-standard shapes and multiple materials with different absorption and dispersion properties, such as the interior of a car (which is the setting where INFER is tested).

INFER predicts both a complex attenuation factor (with independent predictions for magnitude and phase) as well as a directional retransmission factor. The final transfer function is computed by casting 32 rays from each source and accumulating the effects of each ray across 64 points, until we arrive to the query point. Once the transfer function is accumulated, a loss with 6 terms is computed that ensures the predictions of the network are physically realistic.

Evaluations on both simulated and recorded data such as a model cabin and an actual cabin of a Tesla Model X car show that INFER model the transfer function significantly better than other neural network methods which model the transfer function in the time domain.

**Strengths:**

* INFER applies machine learning techniques to an interesting domain (transfer function estimation in complex environments), which would be difficult to solve with traditional signal processing techniques.
* The manuscript contains a useful primer on the physics of audio propagation, making the paper accessible to non-acoustics-experts.
* The paper is really well motivated, particularly around the need for frequency domain modelling for the neural network.

**Weaknesses:**

* **W1**: While the motivation, related works, and acoustics primer are really well developed; the machine learning aspects, the ray marching strategy, and the evaluation are missing important details and ablations (see questions below for more details).

* **W2**: There is nothing in INFER that limits it to car cabins. In principle, INFER should be able to approximate the transfer function in any environment. Consequently, the contribution of this work could be strengthened if there was an evaluation on other domains (eg. normal rooms, open environments, other complex environments). At the very least, a discussion of what environments INFER is ideal for should be added.

**Questions:**

### Machine Learning Questions

* **Q1**: What is the training set of INFER? Is it a random split of the datasets presented in sec 5.1?
* **Q2**: How many parameters does each network (attenuation, retransmission) have?
* **Q3**: What layer from the attenuation network is used for conditioning the retransmission network?
* **Q4**: [less important] Could you obtain better results with a convolutional neural network (possibly with parallel branches of multiple kernels)? After all, temporal and frequency features in a STFT are highly locally correlated.
* **Q5** [less important] A diagram of both networks would help better understand the ML contribution.

------

### Ray Marching Questions

* **Q6**: Can you provide further details on the ray marching setting? How do you ensure all the rays converge on p? Do you produce new rays at intermediate points? Do you perform any culling?. Overall, ray marching is only briefly described in the manuscript, but it deserves a more thorough explanation since it is a key part of INFER.

* **Q7**: What are the consequences of increasing/decreasing the number of rays/points for the ray marching?

-----

### Evaluation Questions

* **Q8**: What is the performance with the simulated (COMSOL) module? I was expecting to find those results in Table 1.
* **Q9**: What is the spread of the prediction errors across different 3d positions and frequencies for each method. Perhaps a heat map plotting {position-across-line x frequency x amplitude/phase error} would be helpful to illustrate the behaviour of INFER?
* **Q10**: What exactly is being reported in Tables 1 & 2? I assume it's the mean absolute error but this needs to be explicitly specified.
* **Q11**: How were the T60 and EDT errors computed? I assume directly from the resulting transfer function, but this needs to be explicitly specified.
* **Q12**: For Table 2, the frequency breakdowns were computed with the simulated, buck or Model X datasets?
* **Q13**: Can you provide errors for a higher range of frequencies (for instance third-octave)?
* **Q14**: The loss has many sub terms and the contribution of each one is not well understood. How were the relative loss weights established? Could you ablate the contribution of each term? For instance you could evaluated INFER with $\\{\lambda_\text{spec}, \lambda_\text{mag}, \lambda_\text{phase}, \dots \\} = 0$ and record the decrease in performance for each setting.
* **Q15**: What are the exact terms in $\lambda_\text{aux}$? This should at least be defined in the appendix.
* **Q16**: How were the $\omega$'s determined? What frequencies were de-emphasised and why?
* **Q17**: [less important]: I would suggest a user study where some participants (ideally $N>15$) listen to the sounds in the car cabin convolved, and then the same sounds convolved with different transfer functions (INFER, INRAS, AVR, NAF). Participants would then rate their subjective impressions of which one is closer to the baseline. I hypothesise INFER would do better than other baselines, but more importantly such an experiment would tell us how far we are from an "ideal" transfer function estimation method.
* **Q18**: Will the training and evaluation datasets be released? Sec 8 mentions "demo" datasets.

-----

### Nitpicks (do not affect rating, no need to follow up on rebuttal)

* **N1**: Lines 70-75 discuss related work, and indeed the same points are repeated in sec 2.1. I would suggest removing them or heavily summarising them.
* **N2**: $\Omega$ is undefined in eq. (3). I assume it is the volume being modelled, but this needs to be explicitly specified.
* **N3**: Sec 3.3 uses $G(x,x')$ but the rest of the manuscript seems to refer to the same concept as $\delta(x)$. I would stick to one notation to ease readability.
* **N4**: The bibliography needs cleaning up: some surnames are in all-caps, the same conference is sometimes in title-case, sometimes not.
* **N5**: What is TOF in Fig1?
* **N6**: In 4.2 $\hat{n}$ has unit length right? If so I would explicitly mention this.
* **N7**: I would suggest using another symbol for the smoothing filter around line 328. $\mathcal{S}$ is being used to denote the retransmission function.
* **N8**: In line 403 the model outputs $\sigma, \beta$ and $\mathcal{S}$ rather than $H$ correct? H is computed using equation (7) afterwards, isn't it?. If so line 403 needs to be rephrased.

---

> ### Author Response · Authors · 2025-11-20
> **Additional Room Scale Experiments and Ablation Studies**
>
> We sincerely thank Reviewer fi6v for the detailed feedback. We appreciate the recognition of our work's motivation, accessibility, and application. We have conducted substantial additional experiments and clarifications to address all concerns. Below we provide comprehensive responses to each question and weakness, with new experimental results.
>
> # **W1: Missing Details in ML Aspects, Ray Marching, and Evaluation**
>
> We acknowledge this concern and provide comprehensive details below. We have also added substantial material to the appendix and have restructured the main paper to improve clarity.
>
> **Key additions:**
>
> - Extensive ablation studies (Q14)
> - Additional evaluations and visualizations (Q9)
> - Complete ML architecture details (Q1–Q5)
> - Thorough ray marching explanation with ablations (Q6–Q7)
> - Detailed metric definitions (Q10–Q11)
>
> All details are provided in the responses below.
>
> # **W2: Evaluation on Other Domains and Generalizability**
>
> We have added **MeshRIR, RAF-Furnished and RAF-Empty evaluations**. INFER achieves the best or tied-best performance across these settings. We also show that incorporating KK consistency regularizer improves performance compared to and the KK-free ablation. This supports the claim that INFER is a general frequency-response-field framework, with cars as one high-value application domain. A table is as follows -
>
> ## Table 2: Evaluation on room-scale environments.
>
> | Method | MeshRIR |  |  |  |  |  | RAF-Furnished |  |  |  |  |  | RAF-Empty |  |  |  |  |  |
> |--------|---------|------|------|------|------|------|---------|------|------|------|------|------|---------|------|------|------|------|------|
> |  | Phase | Amp. | Env. | T60 | C50 | EDT | Phase | Amp. | Env. | T60 | C50 | EDT | Phase | Amp. | Env. | T60 | C50 | EDT |
> | AAC-nearest | 1.47 | 0.91 | 1.40 | 8.6 | 2.20 | 58.8 | 1.60 | 1.09 | 4.83 | 13.0 | 3.41 | 73.5 | 1.60 | 1.09 | 4.83 | 13.0 | 3.41 | 73.3 |
> | AAC-linear | 1.44 | 0.89 | 1.42 | 8.2 | 2.29 | 58.9 | 1.60 | 0.99 | 3.81 | 12.4 | 3.65 | 90.2 | 1.59 | 1.10 | 5.22 | 13.1 | 3.25 | 71.5 |
> | Opus-nearest | 1.45 | 0.72 | 1.37 | 5.2 | 1.26 | 35.7 | 1.60 | 1.19 | 5.35 | 14.4 | 3.78 | 80.3 | 1.59 | 1.16 | 4.58 | 13.3 | 4.25 | 100.6 |
> | Opus-linear | 1.43 | 0.69 | 1.37 | 6.9 | 1.83 | 49.3 | 1.60 | 1.47 | 5.74 | 13.1 | 3.55 | 77.8 | 1.59 | 0.95 | 4.26 | 12.7 | 3.94 | 95.5 |
> | NAF | 1.61 | 0.64 | 1.59 | 4.2 | 1.25 | 39.0 | 1.62 | 0.93 | 5.34 | 7.1 | **0.98** | **20.6** | 1.62 | 0.85 | 4.67 | 8.0 | 1.22 | 26.3 |
> | INRAS | 1.61 | 0.77 | 1.85 | 3.4 | 1.47 | 40.7 | 1.62 | 0.96 | 6.43 | 6.9 | 1.08 | 21.4 | 1.62 | 0.88 | 4.72 | 7.6 | 1.21 | 25.8 |
> | AVR | 1.48 | 0.54 | **1.15** | 3.9 | 0.92 | 35.1 | **1.58** | 0.28 | 5.79 | 6.6 | 1.12 | 22.88 | **1.58** | 0.29 | 5.16 | 6.3 | 1.18 | 24.3 |
> | INFER (without KK) | 1.224 | 0.26 | 7.40 | 2.8 | 0.57 | 127.1 | **1.58** | **0.2337** | 5.33 | 6.35 | 1.15 | 22.56 | **1.58** | **0.23** | 4.73 | 6.0 | 1.07 | 23.1 |
> | INFER | **1.194** | **0.24** | 7.34 | **3.14** | **0.50** | **12.45** | **1.58** | **0.2197** | **5.40** | **6.34** | **1.08** | **22.17** | **1.58** | **0.23** | **4.76** | **6.3** | **1.11** | **23.5** |
>
> These results have now been added to Table 2 in  Sec. 5.3
>
> ## Questions
>
> **Q1: What is the training set of INFER? Is it a random split of the datasets in Sec. 5.1?**
>
> INFER is trained independently in each dataset mentioned in  Sec. 5.1 .Each dataset is divided in 75:15:10 for training set, testing set and validation set. These and additional details have been added to Appendix Sec A.2
>
> **Q2: How many parameters does each network (attenuation, retransmission) have?**
>
> The Attenuation Networks contains 670,978 parameters (76,416 for encoder (3 layers 128 neurons) + 594,562 decoder (3 layers 128 neurons)). The Retransmission Network has 2,840,578 parameters ((3 layers 512 neurons)). Thus the total parameters are 3,511,556 parameters (~3.51M). These details have been added to Appendix Sec A.2
>
> **Q3: What layer from the attenuation network is used for conditioning the retransmission network?**
>
> The retransmission (signal) network is conditioned using the output of the Attenuation Encoder, after ReLU activation. We have added a diagram for the network architecture in the Appendix (Figure 5 Append Sec A.2)
>
> **Q4 Would a convolutional network work better, given local STFT correlations?**
>
> **A4:** This is an interesting direction. Our current design operates on *coordinate inputs* (positions, directions, frequency index) rather than STFT images; thus convolution is less natural than for grid-structured inputs. We chose coordinate-MLPs to align with implicit neural representation literature. Exploring feasibility of convolutional or kernelized variants  will be discussed as future work.
>
> **Q5 A diagram of both networks would help.**
>
> **A5:** We have added a diagram for the network architecture in the Appendix (Figure 5 Append Sec A.2)

---

> > ### Author Response · Authors · 2025-11-20
> >
> > ### Ray marching questions (Q6–Q7)
> >
> > **Q6: Further details on the ray-marching setting? How do rays “converge” on p? Are new rays produced? Any culling?**
> >
> > **A6:** Rays are cast *from the receiver position* pr over a fixed azimuth–elevation grid; they do not need to converge to an intermediate point p. Instead, each ray samples points along its path and accumulates attenuation and retransmission toward the receiver. At each step along a ray, we query the neural field at the current spatial coordinate, integrating complex transmittance and directional spectra as in Eq. (5–7). We do not spawn secondary rays; multiple-bounce behavior is implicitly captured by the learned attenuation and retransmission fields. Rays are culled early when the accumulated amplitude falls below a small threshold, which we will state explicitly. We have added a new section to the Appendix (Sec. A.4) explaining this in detail.
> >
> > **Q7: Consequences of increasing/decreasing the number of rays/points?**
> >
> > **A7:** We have run an ablation varying the number of rays per receiver and the number of samples per ray. Decreasing either quantity significantly below the default (32 rays × 64 samples) degrades performance. Increasing them beyond the default substantially increased compute cost beyond GPU memory size. We have added these results as a part of our ablation table (Table 3)
> >
> > | Study Objectives | Variation | Phase | Amp. | Env. | T60 | EDT |
> > |------------------|-----------|-------|------|------|-----|-----|
> > | Sampling Parameters | 32 × 16 rays, 64 points | 0.98 | 0.43 | 4.2 | 13.6 | 10.2 |
> > |  | 48 × 24 rays, 64 points | 0.91 | 0.24 | 1.9 | 10.06 | 6.1 |
> > |  | 64 × 32 rays, 64 points | 0.5 | 0.12  |  0.95  | 9.8 | **2.6** |
> > |  | 64 × 32 rays, 40 points | 1.13 | 0.32 | 2.3 | **7.0** | 6.9 |
> > |  | 64 × 32 rays, 70 points | **0.48** | **0.11** | **0.91** | 7.8 | 2.8 |
> >
> > ### Evaluation questions (Q8–Q18)
> >
> > **Q8: What is the performance on the simulated COMSOL module?**
> >
> > **A8:** COMSOL results have been added to a new Appendix section (A.3 Additional Evaluations, Table 7) due to page limit constraints. The evaluation underlines INFER’s superior performance on just in Real world datasets but also simulation based datasets.
> >
> > | Method | Amp | Ang | Spec | STFT | Ene. | Env. | T60 | EDT |
> > |--------|------|------|------|------|------|-------|-------|--------|
> > | INRAS | 1.28 | 1.60 | 2.18 | 4.11 | 2.95 | 12.88 | 14.6 | 51.58 |
> > | NAF | 1.53 | 1.61 | 2.76 | 3.53 | 5.40 | 19.41 | 29.09 | 102.88 |
> > | AVR | 0.81 | 1.60 | 2.07 | 5.01 | 3.12 | 17.08 | 26.36 | 35.14 |
> > | INFER | **0.78** | **1.60** | 2.42 | **3.10** | **2.75** | **7.2** | **12.9** | **29.4** |
> >
> > **Q9: Error spread across positions and frequencies; heatmap visualization.**
> >
> > **A9:** We agree that spatial–spectral error structure is informative. We have generated heatmaps of amplitude and phase errors as a function of receiver position. These plots show that INFER achieves consistently low and spatially uniform errors, whereas baseline methods exhibit localized error concentrations and phase discontinuities. This plot has been added to new Appendix section (A.3 Additional Evaluations, Figure 5)
> >
> > **Q10: What exactly is reported in Tables 1 and 2?**
> >
> > **A10:** Tables 1 & 2(now Table 1 & 4) report **Mean Absolute Error (MAE)** for all metrics. We have now specifically mentioned this in the Table caption.  We have updated the metrics definitions to give more details in Appendix Sec. A.9 .
> >
> > **Q11: How are T60 and EDT errors computed?**
> >
> > **A11:** Yes, T60 and EDT are computed directly from the reconstructed impulse responses. We compute these metrics from both predicted and ground truth impulse responses (converted from frequency domain via IFFT for our method), then report the Mean Absolute Error between them in milliseconds. We have added explicit definitions for all metrics including T60 and EDT in **Appendix Sec A.9**
> >
> > **Q12: For Table 2, which dataset is used for the frequency breakdowns?**
> >
> > **A12:** Table 2’s per-band breakdown currently uses the Buck dataset. In the revision, we will state this explicitly in the caption
> >
> > **Q13: Errors for a higher range of frequencies (e.g., third-octave).**
> >
> > **A13:** We have extended the analysis to third-octave bands in Appendix Sec A.3 Table 8. INFER achieves the lowest magnitude and phase error in nearly every band, including both low-frequency (100–400 Hz) and high-frequency ( greater than  2 kHz) regions.

---

> ### Author Response · Authors · 2025-11-20
>
> **Q14 Loss Weights and Ablation:**
>
> A14: We determined the loss weights via grid search to prioritize frequency-domain reconstruction while preserving physical consistency. Comprehensive ablations (Table 3, p. 10, L486–510) evaluate INFER with each loss term individually removed ($\{\lambda_{\text{spec}}, \lambda_{\text{mag}}, \lambda_{\text{phase}}, \ldots\} = 0$ ), showing consistent degradation and validating our multi-objective loss design.
>
> | Study Objectives | Variation | Phase | Amp. | Env. | T60 | EDT |
> |------------------|-----------|-------|------|------|-----|-----|
> | Loss Component | w/o mag loss | 0.74 | 0.18 | 1.6 | 8.9 | 7.1 |
> |  | w/o phase loss | 0.98 | 0.2 | 1.8 | 7.8 | 6.2 |
> |  | w/o energy loss | 0.5 | **0.12** | 0.99 | 24.0 | 3.5 |
> |  | w/o kk loss | 0.77 | 0.18 | 1.7 | 9.8 | 7.3 |
> |  | w/o stft loss | 0.64 | 0.15 | 1.4 | 7.0 | 4.4 |
> |  | w/o spec loss | 1.44 | 0.25 | 2.6 | **2.4** | 2.7 |
> |  | w/o env loss | 0.57 | 0.13 | 1.2 | 5.7 | 3.9 |
> |  | w/o frequency weighting | 0.55 | 0.13 | 1.2 | 2.8 | 3.4 |
> |  | w/ all loss components | **0.48** | **0.12** | **0.95** | 9.8 | **2.6** |
>
> **Q15: Terms in $L_{\text{aux}}$**
>
> A15: The auxiliary loss term $L_{\text{aux}}$ is mentioned in Appendix A.5 (Loss Functions and Weights) on page 15, lines 837-843. $L_{\text{aux}}$ comprises two components: $L_{\text{aux}} = \lambda_{\text{STFT}} L_{\text{MR-STFT}} + \lambda_{\text{energy}} L_{\text{energy}}$
>
> where: $L_{\text{MR-STFT}}$ is the multi-resolution STFT loss and $L_{\text{energy}}$ is the energy loss measuring cumulative energy deviation in the frequency domain
>
> **Q16: Frequency Weighting $w(f)$:**
>
> A16: The frequency weights $w(f)$ are explicitly defined in Appendix A.5 (Loss Functions and Weights) on page 15, lines 837-843. We determined these weights based on psychoacoustic principles and hardware characteristics of our audio system. The ablation study (Table 3, "w/o frequency weighting") shows this weighting provides consistent improvements across all metrics.
>
> **Q17 Suggestion of a perceptual listening test.**
>
> **A17:** A listening study comparing INFER and baselines would indeed be valuable and is planned as follow-up work with our industry partners. Due to space and time constraints, we focus this paper on objective metrics aligned with practice (frequency response and room-acoustic descriptors).
>
> **Q18: Will the training and evaluation datasets be released?**
>
> **A18:** We have released the demo dataset on our project page: [https://anonymous1415510-spec.github.io](https://anonymous1415510-spec.github.io/). The full dataset would be released subject to company’s approval.
>
>
> ## Response to Nitpicks
>
> ### **N1: Lines 70-75 Repetition with Section 2.1**
>
> **We agree.** We will remove lines 70-75 and consolidate all related work discussion in Section 2.1 to avoid redundancy.
>
> ### **N2: Ω Undefined in Eq. (3)**
>
> **Corrected.** Ω is the spatial volume being modeled. We will add: "where Ω ⊂ ℝ³ denotes the volume of the environment."
>
> ### **N3: Notation Inconsistency**
>
> **Thank you for catching this.** We clarify that **G(x, x')** (Green's function in Sec. 3.3) and **δ(f, x)** (local complex attenuation in Sec. 3.4) represent related but distinct concepts. **G(x, x')**: Complete propagation kernel from point x' to x whereas **δ(f, x)**: Local material property at point x whose path-integrated effects determine G. To improve readability, we have added the following clarification after Equation (3):
>
> > "In practice, for lossy media, G(x, x') cannot be computed analytically and depends on the spatial distribution of material properties along the propagation path. In Section 3.4, we introduce the local complex attenuation field δ(f, x), whose path-integrated effects determine the effective Green's function between any two points."
> >
>
> This bridges the conceptual gap between the two sections and maintains mathematical precision while improving clarity.
>
> ### **N4: Bibliography Formatting**
>
> We have cleaned up surname capitalizations , conference names.
>
> ### **N5: TOF Definition in Fig. 1**
>
> **TOF = Time-of-Flight** (propagation delay). We have added this to the figure caption: "TOF: time-of-flight delay due to propagation distance."
>
> ### **N6: n̂ Unit Length in Section 4.2**
>
> **Confirmed.** All direction vectors are **unit length** (||n̂|| = 1). We will add this to the paper
>
> ### **N7: Symbol S for Smoothing Filter**
>
> We are now using script E for the exponential smoothing filter to distinguish from S(f,p,n̂) the retransmission function.
>
> ### **N8: Line 403 Output Clarification**
> You're right—the model outputs δ[f] and S[f], then H(f) is computed via Equation (7). We have rephrase line 375 to: "The model outputs the corresponding complex attenuation δ[f ] ∈ CT and directional spectrum S[f ] ∈ CT at that query point, from which the frequency response H[f ] is rendered using Eq.7."
>
> We hope these additional experiments and clarifications address your concerns and motivate an upward revision of the score.

---

> ### Author Response · Authors · 2025-11-25
> **Follow-up on author response**
>
> Dear Reviewer fi6v,
> Thanks again for your feedback regarding our paper. We have responded to your initial comments/questions and have incorporated them accordingly into our revised manuscript.
>
> We would greatly appreciate you taking a moment to review our response, reassess our revised manuscript from all aspects, share your thoughts, and update your score accordingly. We sincerely believe that our paper has been strengthened a lot thanks to the feedback. We look forward to hearing from you.

---

### Author Response · Authors · 2025-12-01
**INFER: Post‑Rebuttal Summary for AC**

Dear Area Chair,

 In light of the ICLR 2026 score-freeze decision, I would like to briefly summarize how the revised manuscript and rebuttal address the main concerns raised in the reviews, and why I believe the paper now meets the bar for acceptance even though the reviewer scores cannot be updated.

 1. **Generalization beyond car cabins / “only for cars” concern (all reviewers).**
   - In addition to COMSOL, BUCK, and Tesla, we now extensively evaluate on public room-scale datasets (MeshRIR, RAF-Furnished, RAF-Empty), where INFER achieves best or tied-best performance across phase, amplitude, envelope, and acoustic metrics compared to NAF, INRAS, AVR and interpolation/codec baselines. This directly addresses the concern that the approach might be limited to car cabins.

 2. **Missing technical and experimental details (Reviewer fi6v, dpmJ).**
   - We have added in-depth details of the training setup (dataset splits, architecture sizes for attenuation and retransmission networks, ray-marching procedure, and renderer configuration) and moved them into an explicit appendix section with a new architecture figure.
   - We clarified all evaluation metrics (including T60 and EDT), how they are computed from the reconstructed impulse responses, and explicitly stated that Tables 1–4 report mean absolute error.

 3. **Need for ablations and understanding of what drives gains (fi6v, dpmJ).**
   - We added a comprehensive ablation table over all loss components (magnitude, phase, spectral, envelope, KK, STFT, energy), sampling parameters (number of rays and points), and training-data sparsity. These show that removing any of these components consistently degrades magnitude/phase and room-acoustic metrics, and that performance degrades smoothly as data become sparser, supporting both the design and robustness claims.

 4. **Novelty and role of KK regularization and hardware/perceptual weighting (dpmJ).**
   - We clarified that the contribution is not simply “frequency-domain processing”, but a continuous neural **frequency response field** that jointly learns frequency-dependent attenuation and directional retransmission, with an explicit KK-consistency regularizer and psychoacoustic/hardware-aware spectral weighting.
   - New ablations show that removing KK or the frequency weighting worsens both magnitude and phase errors and leads to less physically plausible amplitude–phase behavior, supporting the interpretability and fidelity claims.

 5. **Baseline fairness and strength (dpmJ, qS5V).**
   - All baselines (NAF, INRAS, AVR and interpolation/codec methods) are re-implemented with the same backbone architecture, data splits, and training budget, and we verified their behavior on public datasets to match reported trends. INFER’s gains are therefore on top of strong and carefully controlled baselines rather than from additional data or capacity.

 Overall, the reviews consistently acknowledge that the problem is important, the physics grounding is solid, and the initial results are strong; the low scores mainly reflected missing details and limited experimental scope at submission time. The revised manuscript and rebuttal now add substantial new experiments and clarifications that directly target those weaknesses, but the frozen-score policy prevents reviewers from updating their ratings.

 Given that the core technical and empirical concerns have been addressed with concrete new evidence, and that the method now demonstrates both strong performance and generality beyond car cabins, I respectfully ask that you base your decision primarily on the revised version and rebuttal rather than the initial numerical scores.

---

### Meta-Review · Area_Chair_fdcF · 2026-01-07

**Summary:**

This paper received mixed reviews. The main concerns raised in the reviews are:
1. missing technical details (`fi6v`, `dpmJ`).
2. limited technical innovation (`dpmJ`).
3. limited evaluation in car cabin environments (`fi6v`, `dpmJ`, `qS5V`)

The initial version of the submission had many issues regarding both the presentation and the experiment design, as identified by the reviewers. I agree the revised manuscript is now much stronger with additional technical details and new experiments on room-scale environments. Although the results on generic room-scale environments seem slightly better than the existing methods, the authors keep the primary claims on confined environments like car cabins, which still limits its impacts.

Overall, the proposed method seems quite promising, even on generic environments. However, the current presentation and experiments appear overly narrow, despite substantial revisions. I do think the submission is ready for publication at ICLR at its current state. The authors may consider resubmitting the paper to a more specifically engineering venue or reconsider the potential impacts of the method and revise the paper for a general AI audience.

**Reviewer Concerns:**

1. Concern #1 is partially addressed by the detailed responses, but it is difficult to determine whether all questions have been fully answered.
2. Concern #2 is mostly addressed with the clarifications on the differences from existing approaches.
3. Concern #3 is only partially addressed by the preliminary experiments on room-scale environments. I believe the results can be further improved if this were the targeted application scenarios from the beginning. However, the current results do not appear adequate in both performance improvement and completeness to support a full claim on generic scenes.

**Reviewer Scores:**

1. Reviewer `fi6v` (2->2+): the reviewer might increase the rating, but it is unlikely that they will upgrade it to a positive score.
2. Reviewer `dpmJ` (2->2+): the reviewer might increase the rating, but it is unlikely that they will upgrade it to a positive score.
3. Reviewer `qS5V` (6->?): it is difficult to predict how this reviewer would update the rating if they had participated fully in the discussion.

---

### Decision · Program_Chairs · 2026-01-26

Reject